# Position: Out of Control - Why Alignment Needs Formal Control Theory

## Abstract

This position paper argues that formal optimal control theory should be central to AI alignment research, offering a distinct perspective from prevailing existing AI safety and security approaches. While recent work in AI safety and mechanistic interpretability has advanced formal methods for alignment, they often fall short of the generalisation required of other control frameworks required of other technologies. There is also a lack of research into how to render different alignment/control protocols interoperable. We argue that by recasting alignment through principles of formal optimal control and framing alignment in terms of hierarchical stack from physical to sociotechnical layers according to which controls may be applied we can develop a better understanding of the potential and limitations for controlling frontier models and agentic AI systems. To this end, we introduce an *Alignment Control Stack* and formal methods to address these challenges and demonstrate their utility in simulated experiments. We argue that such analysis is also key to the assurances that will be needed by governments and regulators in order to see AI technologies sustainability benefit the community. Our position is that doing so will bridge the well-established and empirically validated methods of optimal control with practical deployment considerations to create a more comprehensive alignment framework, enhancing how we approach safety and reliability for advanced AI systems.

## 1 Introduction

Current AI alignment research is a vast enterprise incorporating technical, engineering, theoretical and sociotechnical approaches to governance of AI systems. In particular, AI safety and security research has explored various avenues, including empirical evaluations of AI control protocols [1, 2], assessing subversion capabilities [3], developing adaptive deployment strategies for untrusted models [4], and mitigating specific threats like steganographic collusion [5]. These works, along with research into mechanistic interpretability[6, 7, 8] and alignment and AI safety more generally, aim to make AI systems safer and more understandable. While these contributions are valuable, this paper posits that there are two systematic gaps in the AI alignment and control literature requiring attention: (a) the *formalisation problem*: much alignment research - even technical research - is often confined to empirical observations about specific models and lacks formalisation common in control literature. This makes it difficult to generalise results or compare techniques in rigorous ways; (b) the *alignment coordination problem* - alignment as a concept covers theoretical and applied engineering disciplines all the way to socio-technical fields. There is a lack of a clear layered control taxonomy to help organise *where* in the AI stack (from physical circuitry to the socio-technical) controls are directed at, how each layer of control interacts *vertically* and *horizontally* (when multiple systems are involved).

**Position**: To address these gaps, our position argues for:

Submitted to 39th Conference on Neural Information Processing Systems (NeurIPS 2025). Do not distribute.

1. **Alignment Control Stacks to address the coordination problem**: that alignment reearch ought to adopt a stack-based model that clearly specifies where in the overall AI stack alignment and control are directed.

2. **Control theory to solve the formalisation problem**: that alignment research ought to integrate better principles of formal control theory - *even where a layer or element of the stack is difficult to formalise or uncertain*, control theory provides useful tools for estimating the uncertain degree of control which is useful in risk assessments.

Combining a structured hierarchical taxonomy of alignment with formal control methods we believe will assist in facilitating a more coherent approach to alignment research and enable greater clarity regarding interoperability of control proposals. To this end our Position paper sets out: (a) an example of the technical control stack denoted the *Alignment Control Stack* - a hierarchical taxonomy that indicates the level at which measurement and control are directed; and (b) examples of how formal control frameworks can enhance emerging AI control protocols and research (including via replicating recent AI control results via toy simulations to illustrate the benefits of a control theory approach.

## 1.1 Control is Ubiquitous

Formal control theory, encompassing concepts like system modelling (state-space representations), controller synthesis, stability analysis (e.g., Lyapunov theory), robust control (e.g., $\mathcal{H}_\infty$), reachability analysis, and game-theoretic formulations for adversarial settings, provides a powerful toolkit for analysing and ensuring the safe behaviour of complex systems [9, 10]. The principles of control theory are not limited to a single aspect of AI system design or operation. Rather, they can be conceptualised as a full stack of potential interventions and measurement points, spanning from the lowest levels of physical hardware to the highest levels of societal interaction and emergent multi-agent phenomena. This holistic view recognises that AI systems are complex, multi-layered entities, and opportunities for enhancing safety, reliability, and alignment through control exist at each layer.

For instance, at the physical infrastructure layer, control methods might involve thermal management or workload scheduling - essentially controlling compute capability. Moving up the stack, system software controls may include resource quotas or sandboxing. At the AI framework and model architecture layers, controls like learning rate scheduling, regularization, or even architectural choices (e.g., pruning, quantization) are common. The training process itself is a complex control problem, involving data pipeline management and optimisation dynamics. Importantly, as we move to behavioural outputs, interpretability, and the safety/alignment layers, control encompasses task performance thresholds, adversarial robustness measures, mechanistic interventions derived from circuit-level understanding, reward modelling, and harm prevention filters. Even at the multi-agent and societal levels, control concepts apply through mechanism design, governance protocols, and impact assessments.

## 2 Alignment Taxonomies

Many current AI safety approaches, while innovative and practically oriented, exhibit limitations when viewed through the lens of formal control theory. These limitations often pertain to the lack of formal guarantees, robustness to worst-case scenarios, verifiable generalisation, and insufficient modelling of complex stochastic interactions. There is also a dearth of research into whether and how different alignment and control methods can interoperate. Alignment methods can be broadly classified into several categories: (1) *Preference-based learning*, primarily using Reinforcement Learning from Human Feedback (RLHF) to align models with human preferences by learning to summarise [11], follow instructions [12], or fine-tune from preferences directly [13], often forming the core of assistant development [14]; (2) *Principle-based alignment*, such as Constitutional AI, which uses AI-generated feedback based on a set of rules to guide behaviour [15]; (3) *Theoretical frameworks for value learning*, including cooperative inverse reinforcement learning [16] and scalable reward modelling [17], which aim for more robust value inference; (4) *Empirical safety evaluations and protocol design*, focusing on red teaming to discover harms [18], developing AI control protocols against subversion [1], and adaptive deployment strategies for untrusted models [4]; (5) *Understanding and mitigating emergent behaviours and internal model properties*, such as goal misgeneralisation [19, 20], risks from learned optimisation [21] or self-supervised learning leading to SUTRA [22], steganographic

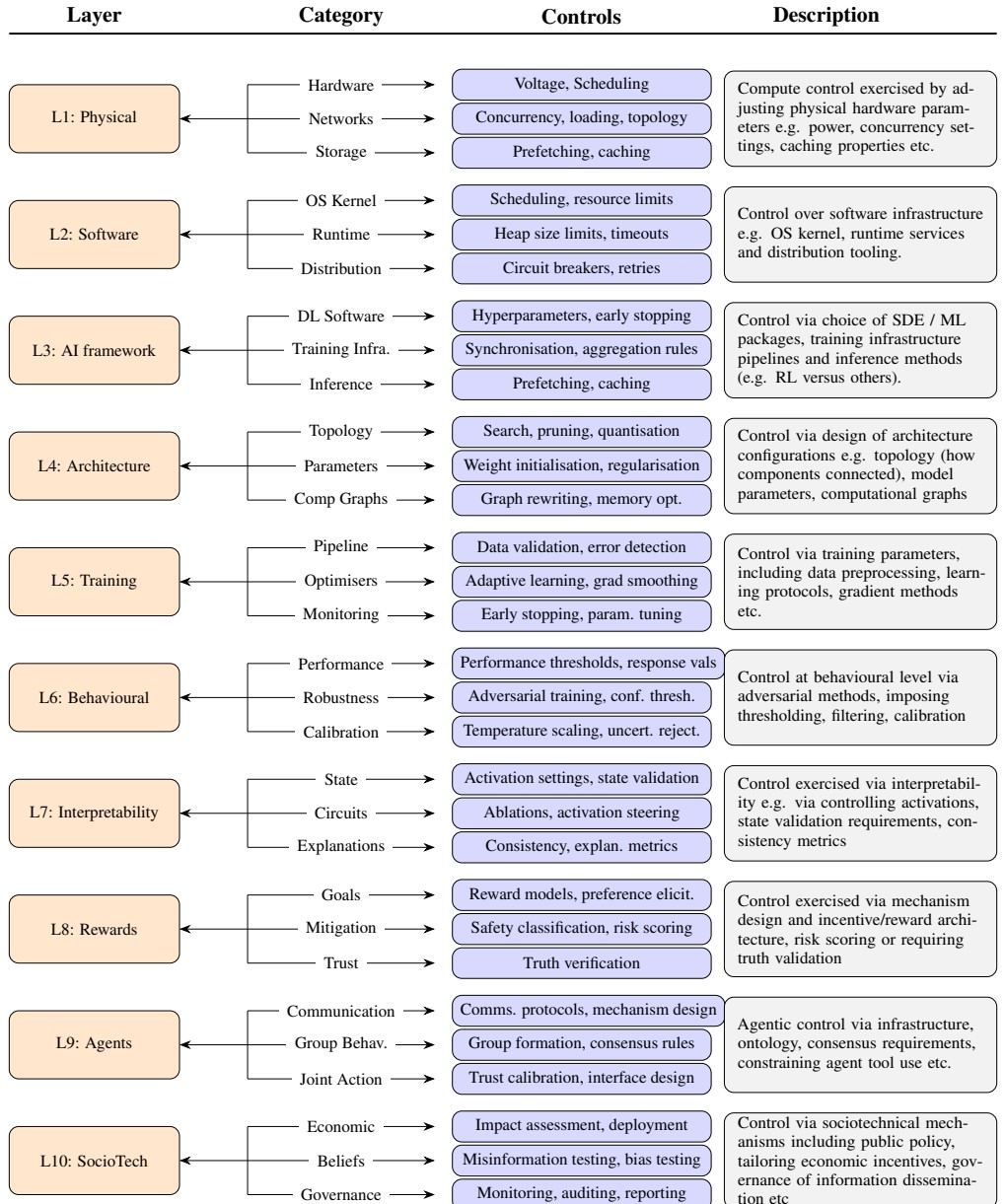

| Layer | Category | Controls | Description |
|-------|----------|----------|-------------|
| L1: Physical | Hardware | Voltage, Scheduling | Compute control exercised by adjusting physical hardware parameters e.g. power, concurrency settings, caching properties etc. |
| | Networks | Concurrency, loading, topology | |
| | Storage | Prefetching, caching | |
| L2: Software | OS Kernel | Scheduling, resource limits | Control over software infrastructure e.g. OS kernel, runtime services and distribution tooling. |
| | Runtime | Heap size limits, timeouts | |
| | Distribution | Circuit breakers, retries | |
| L3: AI framework | DL Software | Hyperparameters, early stopping | Control via choice of SDE / ML packages, training infrastructure pipelines and inference methods (e.g. RL versus others). |
| | Training Infra. | Synchronisation, aggregation rules | |
| | Inference | Prefetching, caching | |
| L4: Architecture | Topology | Search, pruning, quantisation | Control via design of architecture configurations e.g. topology (how components connected), model parameters, computational graphs |
| | Parameters | Weight initialisation, regularisation | |
| | Comp Graphs | Graph rewriting, memory opt. | |
| L5: Training | Pipeline | Data validation, error detection | Control via training parameters, including data preprocessing, learning protocols, gradient methods etc. |
| | Optimisers | Adaptive learning, grad smoothing | |
| | Monitoring | Early stopping, param. tuning | |
| L6: Behavioural | Performance | Performance thresholds, response vals | Control at behavioural level via adversarial methods, imposing thresholding, filtering, calibration |
| | Robustness | Adversarial training, conf. thresh. | |
| | Calibration | Temperature scaling, uncert. reject. | |
| L7: Interpretability | State | Activation settings, state validation | Control exercised via interpretability e.g. via controlling activations, state validation requirements, consistency metrics |
| | Circuits | Ablations, activation steering | |
| | Explanations | Consistency, explan. metrics | |
| L8: Rewards | Goals | Reward models, preference elicit. | Control exercised via mechanism design and incentive/reward architecture, risk scoring or requiring truth validation |
| | Mitigation | Safety classification, risk scoring | |
| | Trust | Truth verification | |
| L9: Agents | Communication | Comms. protocols, mechanism design | Agentic control via infrastructure, ontology, consensus requirements, constraining agent tool use etc. |
| | Group Behav. | Group formation, consensus rules | |
| | Joint Action | Trust calibration, interface design | |
| L10: SocioTech | Economic | Impact assessment, deployment | Control via sociotechnical mechanisms including public policy, tailoring economic incentives, governance of information dissemination etc |
| | Beliefs | Misinformation testing, bias testing | |
| | Governance | Monitoring, auditing, reporting | |

Figure 1: *Aligment Control Stack* (ACS). The ACS details a vertically hierarchical set of layers at which various forms of control may be utilised, ranging from the physical circuitry level (e.g. compute control) through to control via model hyperparameters, model behavioural testing (e.g. adversarial methods) and socio-technical measures (governance).

collusion [5, 23, 24, 25, 26, 27], emergent misalignment from narrow finetuning [28], and discovering latent knowledge [29, 30]; and (6) *Conceptual and mechanistic understanding*, including analyses of power-seeking [31, 32], designing for corrigibility [33], scalable oversight via debate [34, 35, 36], and advancing mechanistic interpretability [8]. While these methods advance AI safety, the direct application of formal optimal control theory, with its emphasis on system modelling, controller synthesis, and robust guarantees, is often implicit or absent, particularly in providing generalisable assurances of safety and alignment across diverse and novel situations.

## 2.1 Gaps in alignment

Current approaches to AI alignment, while varied and rapidly evolving, often exhibit certain limitations that formal control theory is well-suited to address includeing: (1) *Lack of Formal Guarantees*: Many methods, particularly empirical evaluations of safety protocols [37] or red teaming efforts [18, 38], provide evidence of safety under specific conditions but lack formal proofs of safety or alignment across broader operational domains. Formal control's focus on stability analysis, reachability, and robust synthesis may assist in deriving such guarantees. (2) *Specification Gaming and Reward Misspecification*: Defining objectives that perfectly capture human intent is notoriously difficult, leading to models exploiting loopholes in reward functions [39, 40]. While RLHF [41, 12, 42] aims to mitigate this, formal control's emphasis on precise system modelling and objective specification, including handling constraints and uncertainties in cost functionals, can provide a more rigorous framework for objective design. (3) *Generalisation Failures and Emergent Behaviours*: Models can exhibit unexpected or misaligned behaviours when encountering out-of-distribution inputs or through emergent properties not directly trained for. Robust control theory explicitly handles system uncertainty and disturbances, aiming to ensure desired performance even under unmodelled dynamics or unforeseen inputs, which is critical for predicting and managing emergent phenomena. (4) *Scalability of Oversight and Feedback*: Human oversight is a bottleneck for many alignment techniques [17, 43, 44]. While approaches like Constitutional AI [15, 45] or model-written evaluations [46] attempt to scale feedback, formal control offers adaptive control and estimation techniques (e.g., Kalman filtering for belief updates as in Section 4) that can optimize the use of limited supervisory signals. (5) *Interpretability and Intervention*: While mechanistic interpretability seeks to understand model internals and enable targeted interventions like activation steering [47], formal control provides the mathematical tools to model these internal dynamics and design interventions (controllers) that have predictable and verifiable effects on behaviour. (6) *Truthfulness and Deception*: Ensuring AI systems are truthful and do not engage in deception [48] is a complex challenge. Game-theoretic control and POMDPs can model strategic interactions and hidden information, offering avenues to analyse and mitigate deceptive behaviours.

Our position is that greater use of formal control theory and a standardised alignment taxonomy can help towards addressing many of these gaps. AI alignment is not being undertaken in a control theory vacuum - for example many areas such as game theory have been successfully applied to a range of AI scenarios. However, other control formalisms are underutilised. This is particularly the case for stochastic optimal control (see Appendix) which is distinct from, yet complementary to, some proposals for guaranteed safe AI that might rely on purely deductive formal verification from symbolic logic (e.g., [49] for RL shielding). While deterministic methods offer strong deterministic guarantees under fixed assumptions, stochastic control explicitly models and manages uncertainty, noise, and probabilistic outcomes inherent in complex AI systems and their environments (e.g., [50] for adversarial RL). This allows for a more nuanced approach to safety, optimizing for expected outcomes or ensuring high-probability safety bounds rather than absolute guarantees, which may be more practical for inherently stochastic systems like LLMs. It also addresses the main criticism of our proposal namely that formal or 'whitebox' control methods are infeasible by providing a means of framing how we approach the measurement, analysis and control of highly non-linear stochastic systems.

# 3 Alignment Control Stack

To systematically address the formalisation and coordination challenges in alignment research, we propose the use of an Alignment Control Stack, a hierarchical framework decomposing an AI system's lifecycle and operational environment into ten distinct layers. A diagram of the stack is set out in Fig. 1. Each layer presents a means of measurement and control of integrated AI systems. This layered approach facilitates a structured understanding of control points, types, limitations, and interactions, as detailed comprehensively in Table 1 (Appendix D). The idea is that we can specify where different control types—preventive, detective, corrective, and directive—are implemented, their limitations, and how they interrelate, advancing a systematic approach to AI control. The stack in Table 1 decomposes the life-cycle of an AI system into ten vertically-integrated layers, each exposing its own state variables, measurable signals, and admissible control handles:

- At the *lower layers* (1–4) the control layer is essentially physical or software-mechanical; well-understood linear or hybrid control features (voltage regulation, job scheduling, memory bandwidth throttling) dominate.

- *Mid-layers* (5–7) govern the learning dynamics and the representational structure of the model. Their states are high-dimensional and only partially observable; formal stochastic and robust control, often augmented with estimation (Kalman or particle filtering) and adaptive game-theory, becomes critical.

- *Upper layers* (8–10) merge technical control with human-centric governance: value alignment, multi-agent mechanism design, and policy or regulatory feedback loops. Here formal control must interact with economic incentive theory and social-choice constraints, extending the design space beyond purely algorithmic interventions.

Vertical coupling between layers is intrinsic: actions at a low layer propagate upwards as parameter shifts, representation changes, or distributional shifts, while mis-aligned objectives discovered at a high layer must eventually be corrected by interventions that flow back down (e.g. training-data curation, architecture surgery, rate-limiting). Thus a rigorous alignment story demands *hierarchical control synthesis*: layer-local controllers that guarantee local performance while exposing contractable *interface variables* whose behaviour can be used by the next layer up as a well-modelled plant. Such relations can be expressed as guaranteed input–output gains, reachable-set envelopes, or probabilistic performance bounds. The Alignment Control Stack can also be used to illustrate which mathematical or technical assumptions enter at each point. If Layer 6 (behaviour) is modelled as a stochastic map $y_t \sim P_\theta(\,\cdot\,|x_t)$ learned in Layer 5, then any formal guarantee about safe outputs must quantify over both: (i) the stochasticity inherent in $P_\theta$ and (ii) the residual identification error in $\theta$ coming from noisy gradient updates. Layer-wise isolation of these uncertainties enables principled design of robust composite controllers that remain valid even as frontier models push distributional generalisation into ever sparser data regimes. More detail about each layer of the stack and control protocols that would apply are set out in Appendix D.

## 3.1 Alignment Interoperability

Approaching alignment questions using a stack-based approach (such as the Alignment Control Stack) allows us to analyse interoperability and potential interference between different control regimes. For instance, an architectural choice promoting interpretability (Layer 4) might enable more effective reward modelling (Layer 8), while a poorly designed data pipeline (Layer 5) could undermine even sophisticated safety filters at the behavioural output (Layer 6). Formal control theory, by providing a common mathematical language for system dynamics and controller design, allows for the analysis of these inter-layer dependencies, helping to identify synergistic combinations and potential conflicts between different alignment interventions. Without such a structured, formal perspective, efforts to combine various alignment techniques risk becoming ad-hoc, with unpredictable emergent consequences and a lack of generalisable safety assurances. The Alignment Control Stack can be used to facilitate interoperability analysis of interactions both vertically and horizontally:

1. **Vertical integration** describes the hierarchical dependencies where lower layers form the substrate for higher ones, and control actions or system properties at one layer directly influence the state and dynamics of adjacent layers. For example, the stochasticity of the training process (Layer 5), characterized by minibatch gradient noise $\xi_t$, directly affects the evolution of model weights $w_t$, which in turn defines the behavioural output $y_t$ (Layer 6). A control strategy like LQG can then be designed to co-optimize the weight regulation in Layer 5 and the state estimation (via Kalman filtering) in Layer 6, explicitly modelling and managing this vertical coupling. Misaligned objectives or unmitigated risks at higher layers (e.g., societal harm in Layer 10) necessitate feedback controls that propagate downwards, potentially leading to changes in training data (Layer 5) or even model architecture (Layer 4). We illustrate how formal control can facilitate this in Appendix A.1.

2. **Horizontal integration**, on the other hand, refers to the interplay of different control mechanisms or considerations within a single layer, or between parallel processes at the same level of abstraction. For instance, within the Safety and Alignment layer (Layer 8), reward modelling, red teaming, and constitutional AI principles must work in concert. Similarly, in the Multi-Agent layer (Layer 9), communication protocols, incentive structures,

and normative controls are horizontally integrated to shape collective behaviour. Formal control, particularly through multi-objective optimisation or game-theoretic frameworks [51, 52], can provide tools to analyse and design these horizontal interactions to achieve a coherent overall control strategy within and across layers.

## 3.2 Existing Control Theory

So where does formal control fit into our argument? Formal control theory encompasses a diverse toolkit applicable across the Alignment Control Stack. Below we set out examples of where and how formal control theory may apply across the stack:

1. **Offline control** methods, such as optimal controller synthesis via dynamic programming or Pontryagin's Maximum Principle [53, 54], are relevant for designing fixed policies or system parameters pre-deployment, impacting layers like Model Architecture (Layer 4) or the initial setup of the Training Process (Layer 5).

2. **Online control**, by contrast, including adaptive control [55] and Model Predictive Control (MPC) [56], dynamically adjusts control actions based on real-time system measurements and predictions; this is crucial for Behavioural Output (Layer 6) monitoring, Preference and Reward (Layer 8) interventions like adaptive deployment, and even Multi-Agent (Layer 9) coordination.

3. **Deterministic control** assumes perfect knowledge of system dynamics and no noise, useful for initial modelling or when uncertainty is negligible, for instance in some AI Framework (Layer 3) optimisations.

4. **Stochastic control**. AI systems are inherently noisy and operate in uncertain environments, making stochastic control [57, 51] essential for modelling and managing probabilistic behaviours, from stochastic gradient descent in Training (Layer 5) to the probabilistic nature of LLM outputs (Layer 6) and the uncertainty in reward modelling (Layer 8).

5. **Geometric control** [58, 59] focuses on the underlying geometric structures of control systems, offering insights into controllability, observability, and non-linear system behaviour, which could inform the design of more inherently stable or interpretable architectures (Layer 4) or reveal fundamental limitations in controlling complex internal model dynamics (Layer 7).

6. **Robust control** techniques (e.g., $H_\infty$ control [52]) aim to maintain stability and performance despite bounded model uncertainty or disturbances, applicable across many layers to ensure resilience. Hybrid systems control [60] deals with systems exhibiting both continuous and discrete dynamics, relevant for AI agents interacting with discrete event environments or rule-based safety systems (Layer 8).

7. **Learning-based control**, including reinforcement learning [61], bridges control theory with machine learning and is directly applied in RLHF (Layer 8) and could inform adaptive strategies across the stack.

We set out a short primer on the mathematics of control in the Appendix below. In particular, in Appendix D we set out examples of where control formalism may apply at different levels in the stack. The approach is not complete and the use of one formalism over another is highly context-dependent, but our position is that by bridging formalisms in a stack-based approach, we can start to better address the formalisation and coordination gaps present in alignment research. In the following section, we illustrate how such methods can be integrated with the Alignment Control Stack framework to enhance approaches to alignment research.

## 4 Enhancing AI Control via Formal Control

In this section and the Appendix we provide practical examples of how formal control methods and the Alignment Control Stack taxonomy can enhance a number of several recent AI safety and alignment research results on AI control (specifically where it is assumed we may not have fine-grained control over AI systems). The purpose of our analyses is to demonstrate how formal control theory can frame emergent AI control literature, enhance its findings, and identify gaps in generalisability, thereby strengthening the case for its deeper integration into AI alignment research.

**AI Control Evaluations and Safety Cases (Greenblatt et al. [1], Korbak et al. [2]):** These works evaluate protocols (e.g., auditing, monitoring) against intentional subversion. *Control Framing & Enhancement:* The Blue Team's protocol acts as a controller, and the Red Team's LLM as an adversarial system. Our simulations (Figures 2, 4, 5) show that control/game theory allows principled determination of optimal simple adversary strategies (e.g., Red Team's optimal backdoor attempt probability $q^*$) and enables sensitivity analysis of the protocol's parameters (e.g., audit threshold $\tau_{audit}$, untrusted monitor weight $w_U$). Control theory could be used to explore the strategy space and understand robustness. *Generalisability Gaps:* Current evaluations are often empirical point estimates rather than syntheses of provably robust protocols against a formally defined adversary class. Formal system models for LLM dynamics are needed for rigorous guarantees.

*Alignment Control Stack Layers:* Primarily Layer 8 (Preference and Reward) for protocol design and evaluation, and Layer 6 (Behavioural Output) for monitoring and intervention.

**Subversion Strategy Evaluation (Mallen et al. [3]):** This research assesses LLMs' stateless strategic capabilities to subvert protocols. *Control Framing & Enhancement:* The LLM is an agent whose stateless policy is evaluated against an optimal strategy in a game-like environment. Our simulation (Figure 3) uses control theory to derive the optimal stateless subversive policy and its maximal winrate, providing a benchmark using data from the paper. It also quantifies how LLM imperfections (e.g., miscalibrated probability $p$ for attempting a backdoor) degrade performance, highlighting the precise control needed for effective subversion and the strategic cost of statelessness. *Generalisability Gaps:* The stateless assumption simplifies the problem; stateful, adaptive adversaries (requiring POMDP/dynamic game models) are more potent and less explored (as is when stateless LLMs are supplemented by tools and scaffolding infrastructure). Generalisation of observed capabilities to new protocols remains an open question. *Alignment Control Stack Layers:* Primarily Layer 8 (Preference and Reward) for evaluating subversion, and Layer 6 (Behavioural Output) for the LLM's strategic actions.

**Adaptive Deployment of Untrusted LLMs (Wen et al. [4]):** Proposes an adaptive macro-protocol selecting micro-protocols based on an evolving belief (credence) about an untrusted LLM's alignment. *Control Framing & Enhancement:* The problem can be framed as a POMDP, where the macro-protocol is a controller learning the hidden alignment state. Our simulations (Figures 6, 7, 8) visualize Pareto frontiers of safety-utility tradeoffs, show credence evolution as successful state estimation, and perform sensitivity analyses (e.g., to initial credence), revealing robustness and operational limits of the adaptive strategy. *Generalisability Gaps:* The original heuristic policies lack formal optimality or convergence guarantees, especially against strategic manipulation by the untrusted LLM, which would require robust adaptive control or game-theoretic approaches. *Alignment Control Stack Layers:* Primarily Layer 8 (Preference and Reward) for adaptive safety protocols, Layer 6 (Behavioural Output) for micro-protocol selection affecting outputs. The credence update relates to Layer 5 (Training Process) if seen as online learning.

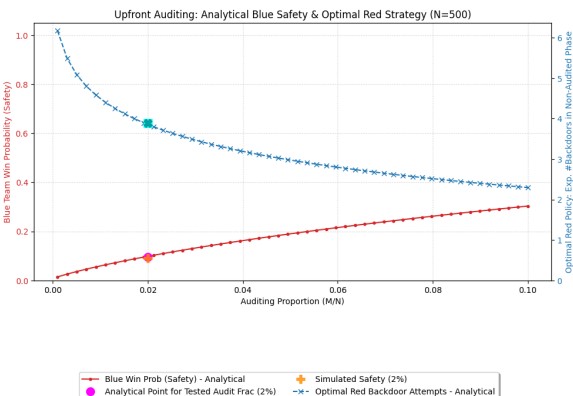

Figure 2: Upfront Auditing: Analytical Blue Team Safety (red line, left $y$-axis) and optimal Red Team expected backdoor attempts in the non-audited phase (blue dashed line, right $y$-axis) as a function of the auditing proportion ($M/N$). For $N = 500$. The magenta dot (on the red curve) marks the analytical $P(\text{Blue Wins})$ at $M/N = 0.02$ (10 audits), $\approx 0.0952$. The orange "+" shows the simulated $P(\text{Blue Wins})$, $\approx 0.0900$, at the same $M/N$ when the Red Team plays its analytical optimum $q^* \approx 0.007953$.

We expand on these examples and a number of others from recent AI control literature in the Appendix. Collectively, formal control theory provides a useful lens for enhancing alignment research at different layers in the Alignement Stack, helping facilitate principled strategy design,

robustness analysis against defined adversaries, understanding of system dynamics and tradeoffs, and formalising guarantees—all crucial for building trustworthy AI systems.

# 5 Alternative Views

While we advocate for formal control theory as central to AI alignment, we acknowledge several important counterarguments and alternative perspectives that merit consideration.

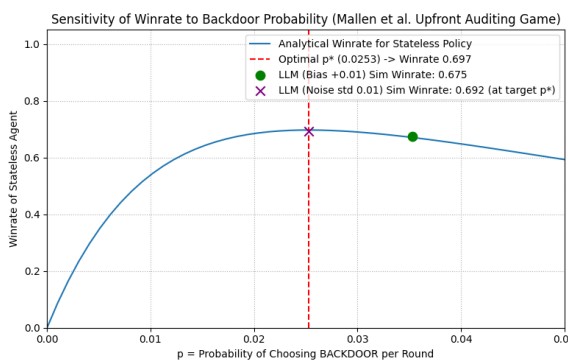

Figure 3: Sensitivity of stateless-agent win-rate to backdoor probability $p$ in Mallen et al.'s upfront-auditing game ($N = 100$, $M = 10$). The blue curve is the analytical win-rate $W(p)$. The red dashed line marks the optimum $p^* \approx 0.0253$ with $W(p^*) \approx 0.697$. The green circle shows a simulated win-rate (0.675) for a mock LLM targeting $p^*$ but executing with a +0.01 bias ($p \approx 0.0353$). The purple cross shows the simulated win-rate (0.692) for a mock LLM targeting $p^*$ but with execution noise (std. dev. 0.01) around its target.

**Complexity, Computability and Intractability Objections** Critics argue that modern AI systems, particularly large language models, are fundamentally too complex and high-dimensional for classical control theory approaches. The state spaces are enormous (billions of parameters), the dynamics are highly nonlinear, and the systems exhibit emergent behaviours that resist formal modelling. Researchers in complexity science argue that the most important AI behaviours emerge from complex interactions that cannot be captured by reductionist control approaches. They advocate for understanding AI systems as complex adaptive systems requiring different analytical tools. Even if formal control approaches are theoretically sound, they may be computationally intractable for real-world AI systems, making them impractical guides for actual deployment.

**Our response**. *While we acknowledge this complexity, we argue that formal control theory provides frameworks for handling uncertainty and complexity, not just tools for fully-specified systems. Stochastic optimal control, robust control, and POMDP approaches are specifically designed for uncertain, high-dimensional systems. Even approximate or hierarchical models can provide valuable insights and safety bounds. Modern control theory, particularly in areas like swarm robotics and complex networks, explicitly deals with emergent behaviours. Multi-agent control, distributed optimisation, and network control theory provide formal frameworks for understanding and steering emergent phenomena. While optimal control can be computationally demanding, many practical control algorithms are designed for real-time implementation. Model Predictive Control, for instance, uses approximate solutions and limited horizons. The goal is not perfect optimality but principled, robust performance within computational constraints.*

**Mechanistic Interpretability Alternative** Some researchers argue that mechanistic interpretability—understanding the internal circuits and representations of AI systems—provides a more direct path to alignment than external control mechanisms. This white-box approach seeks to directly understand and modify internal model dynamics.

**Our response**. *We view mechanistic interpretability as complementary to, not competing with, control theory. Indeed, Layer 7 of our Alignment Control Stack explicitly incorporates interpretability. Control theory can formalise how to use mechanistic insights for interventions, while interpretability can inform better system models for control design.*

**Goodhart's Law and Specification Gaming** Formal control approaches require precise objective specification, but attempts to formalise human values often lead to specification gaming or reward hacking (Goodhart's Law: When a measure becomes a target, it ceases to be a good measure). Critics

argue this makes formal approaches fundamentally unsuited to alignment.

**Our response**. *This critique applies to any optimisation-based approach, not just formal control. However, control theory offers tools specifically designed for robust performance under model uncertainty and adversarial conditions. Techniques like minimax optimisation, robust MPC, and game-theoretic formulations can explicitly account for specification gaming attempts.*

**Empirical Alternatives to Whitebox Controls**  Many practitioners advocate for empirical, iterative approaches to alignment—learning through experimentation, human feedback, and gradual deployment rather than formal analysis due to fundamental constraints on resources and precision. This view emphasises adaptability and learning from real-world deployment over theoretical guarantees.

**Our response**. *We don't oppose empirical approaches but argue they should be grounded in formal frameworks. Control theory itself emphasizes adaptive and learning-based methods (adaptive control, reinforcement learning). The key is having principled frameworks for when and how to adapt, rather than purely ad-hoc responses. While criticisms of limitation of whitebox methods are not without merit, we argue that formal control can still be useful in guiding the development of empirically validated heuristic controls or in providing bounds on uncertainty even when perfect models are elusive.*

# 6   Conclusion: Towards a Control-Theoretic AI Alignment Framework

This position paper argued that formal optimal control theory and the Alignment Control Stack offers set of tools to address to challenges facing alignment research: the *formalisation challenge* - a lack of formalisation which affects how results may be compared and the use of other techniques from related fields; and the *coordination challenge*, how different approaches to alignment and control may be integrated and rendered interoperable. We have introduced the Alignment Control Stack as a means of facilitating focus on where in AI system stacks control and alignment efforts are directed. Via examining formal control and its application to emergent AI control literature, we have highlighted areas where the rigor of control-theoretic approaches—such as formal verification, robust controller synthesis, game-theoretic analysis of adversarial settings, and principled handling of stochasticity—can provide stronger safety guarantees and deeper understanding than current empirical or heuristic methods.

We do not suggest abandoning existing approaches, but rather augmenting and grounding them within a more formal framework. The development of truly aligned and safe AI systems, especially those with agentic capabilities, will require a multidisciplinary effort. We believe that optimal control theory, with its rich history of ensuring safety and performance in complex, critical systems, must be a cornerstone of this endeavor. Future research should focus on:

- Developing tractable but faithful formal models of AI agent dynamics, incorporating insights from mechanistic interpretability to define relevant state variables and transition functions, especially for semantic properties across the AI stack.

- Applying stochastic optimal control and dynamic game theory to model interactions between AI agents and human overseers or other AIs, particularly for analysing emergent deception, collusion, and scheming.

- Designing and formally verifying control protocols that are interoperable vertically and horizontally across the ACS and that offer provable safety and alignment guarantees under well-defined assumptions about AI capabilities and adversarial behaviour.

- Bridging the gap between theoretical control guarantees and practical, scalable implementations for large frontier models.

This path is essential for building societal trust and ensuring that AI technologies are developed and deployed responsibly, with robust assurances of their safety and alignment to human values.

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

# A  Vertical and Horizontal Alignment

## A.1  Vertical Alignment

We illustrate a toy model of how formal control theory can facilitate the study of vertically-integrated alignment below via the interaction of two layers, a deterministic control layer where full control over parameters is possible and a stochastic layer directed at model behaviour such as outputs of LLMs or their behaviour which is non-deterministic.

- **Control Layer 5 – Training Process (deterministic control).**
  *State:* model parameter vector $w_t \in \mathbb{R}^n$. *Control:* learning-rate-scaled update $u_t \in \mathbb{R}^n$ applied by the optimiser. *Nominal dynamics (first-order SGD linearisation):*
  $$w_{t+1} = w_t - u_t + \xi_t, \qquad \xi_t \sim \mathcal{N}(0, \Sigma_\xi), \tag{1}$$
  where $\xi_t$ captures stochastic minibatch gradient noise.

- **Control Layer 6 – Behavioural Output (stochastic).**
  *Output:* scalar task performance or safety metric $y_t \in \mathbb{R}$ derived from the LLM logits at inference time. We adopt the simplest linear-Gaussian surrogate
  $$y_t = c^\top w_t + v_t, \qquad v_t \sim \mathcal{N}(0, \sigma_v^2), \tag{2}$$
  with known read-out vector $c \in \mathbb{R}^n$. The supervision objective is to keep $y_t$ close to a policy-defined target $y^\star$ (e.g. "toxicity score $\tau$").

**Layer 5 – LQR synthesis for weight regulation.** Let $x_t \equiv w_t$ and choose quadratic stage cost: $\ell_t = (w_t - w^\star)^\top Q(w_t - w^\star) + u_t^\top R u_t$, with $Q \succeq 0$, $R \succ 0$. Equation (1) is linear, so the discrete-time Riccati recursion delivers the stationary feedback

$$u_t = -K w_t, \quad K = (R + B^\top P B)^{-1} B^\top P A$$

for $A = I_n$, $B = I_n$ and $P$ the stabilising Riccati solution $P = A^\top P A - A^\top P B(R + B^\top P B)^{-1} B^\top P A + Q$.

**Layer 6 – Kalman filtering and outer regulation.** Because $w_t$ is not directly observed at inference, a Kalman–Bucy filter provides the optimal linear-Gaussian estimate $\hat{w}_t$ from (2). Denote the filter gain $L$. A *secondary* controller (e.g. a safety layer that can mildly post-edit logits) chooses a small correction $\delta_t$ to enforce $\tilde{y}_t = y_t - \delta_t \approx y^\star$ while paying cost $\rho\delta_t^2$. In expectation this is another scalar LQR with state $e_t = y_t - y^\star$:

$$e_{t+1} = c^\top(w_{t+1} - \hat{w}_{t+1}) + \text{noise}; \qquad \delta_t = -k_e\,\hat{e}_t,$$

and gain $k_e$ solves the associated scalar Riccati equation.

## A.2 Vertical interaction as a stacked LQG game

Combine (1) and (2):

$$\begin{bmatrix} w_{t+1} \\ y_t \end{bmatrix} = \underbrace{\begin{bmatrix} I_n \\ c^\top \end{bmatrix}}_{C} w_t + \begin{bmatrix} -I_n \\ 0 \end{bmatrix} u_t + \begin{bmatrix} \xi_t \\ v_t \end{bmatrix}. \tag{3}$$

Define the composite quadratic cost

$$J = \mathbb{E}\sum_{t=0}^{T-1}\Big[(y_t - y^\star)^2 + w_t^\top Q_w w_t + u_t^\top R u_t\Big].$$

Solving the discrete-time *Linear-Quadratic–Gaussian* (LQG) problem for the augmented system yields:

$$\hat{w}_{t+1} = \hat{w}_t - K_w\hat{w}_t + L(y_t - c^\top\hat{w}_t),$$
$$u_t = -K_w\hat{w}_t,$$

with $L$ the steady-state Kalman gain $L = PC^\top(CPC^\top + \sigma_v^2)^{-1}$ and $P$ solving the dual Riccati equation $P = APA^\top - APC^\top(CPC^\top + \sigma_v^2)^{-1}CPA^\top + \Sigma_\xi$.

The separation principle guarantees that this pair $(L, K_w)$ is jointly optimal: Layer 5's weight controller and Layer 6's stochastic estimator interact with provable minimal expected cost even though only $y_t$ is observed and both gradient noise $\xi_t$ and output noise $v_t$ perturb the closed loop.

## A.3 Horizontal Alignment

Horizontal alignment addresses the challenges of ensuring coherent, cooperative, or at least predictably safe interactions between multiple, potentially independent AI systems (or 'stacks'). While vertical alignment concerns the internal coherence of a single stack, horizontal alignment focuses on the dynamics emerging from the interplay of two or more stacks. This is relevant to scenarios involving multi-agent systems, AI-driven markets, collaborative AI tasks, or even potential adversarial interactions between AIs developed by different entities. Formal control theory, particularly through the lens of dynamic game theory, provides the necessary mathematical framework to analyse and design such interactions.

Consider two AI stacks, $A$ and $B$. Each stack $i \in \{A, B\}$ has its own internal state $x_i \in \mathbb{R}^{n_i}$ (potentially encompassing states from various layers of its ACS) and chooses control actions $u_i \in \mathcal{U}_i$. Crucially, their dynamics and objectives are coupled:

- **Coupled Dynamics:** The evolution of each stack's state depends not only on its own actions but also on the state and actions of the other stack.

$$\dot{x}_A = f_A(t, x_A, u_A, x_B, u_B) + \sigma_A(t, x_A, u_A, x_B, u_B)dW_A \tag{4}$$
$$\dot{x}_B = f_B(t, x_B, u_B, x_A, u_A) + \sigma_B(t, x_B, u_B, x_A, u_A)dW_B \tag{5}$$

  where $W_A, W_B$ represent stochastic influences. This coupling can occur at various ACS layers, from shared resource contention (Layer 1/2) to behavioural influence (Layer 6) or communication (Layer 9).

- **Interdependent Objectives:** Each stack seeks to optimize its own objective function $J_i$, which generally depends on the actions of all involved stacks:

$$J_A = \mathbb{E}\left[ g_A(x_A(T), x_B(T)) + \int_0^T L_A(t, x_A, u_A, x_B, u_B) dt \right] \tag{6}$$

$$J_B = \mathbb{E}\left[ g_B(x_B(T), x_A(T)) + \int_0^T L_B(t, x_B, u_B, x_A, u_A) dt \right] \tag{7}$$

The nature of $L_i$ and $g_i$ (derived from Layer 8/9 goals) determines the game's structure (zero-sum, general-sum, cooperative).

**Game-Theoretic Approach** Here the goal is to find strategies (control policies) $\pi_A^*(t, x_A, x_B)$ and $\pi_B^*(t, x_A, x_B)$ such that $u_A = \pi_A^*$ and $u_B = \pi_B^*$ lead to a desirable equilibrium. The most common approach is to find the Nash Equilibrium, where neither player can unilaterally improve its outcome. For dynamic games, this involves finding NE strategies using the HJI equations or equivalents.

Let $V_A(t, x_A, x_B)$ and $V_B(t, x_A, x_B)$ be the value functions for players $A$ and $B$, representing their optimal cost-to-go from state $(x_A, x_B)$ at time $t$. Under a Nash equilibrium, these value functions satisfy a system of coupled, non-linear partial differential equations. Assuming $A$ minimizes $J_A$ and $B$ minimizes $J_B$:

$$-\frac{\partial V_A}{\partial t} = \min_{u_A \in \mathcal{U}_A} \{ L_A(t, x_A, u_A, x_B, \pi_B^*) + \mathcal{L}_A V_A \} \tag{8}$$

$$-\frac{\partial V_B}{\partial t} = \min_{u_B \in \mathcal{U}_B} \{ L_B(t, x_B, u_B, x_A, \pi_A^*) + \mathcal{L}_B V_B \} \tag{9}$$

where $\mathcal{L}_i$ is the second-order differential operator associated with the stochastic dynamics (5) (incorporating both drift $f_i$ and diffusion $\sigma_i$), and $\pi_i^*$ is the optimal strategy for player $i$. Solving this system yields the NE strategies, however considerable resource and tractability difficulties may remain - hence the utility of exploring stochastic control variants to overcome such resource constraints.

The HJI formalism allows us to:

1. *Analyse Potential Outcomes:* Determine if uncoordinated interactions lead to undesirable equilibria (e.g., conflicts, Pareto-suboptimal outcomes like the Prisoner's Dilemma, or harmful collusion against humans).

2. *Design Mechanisms for Alignment:* Act as a mechanism designer (Layer 9/10) to shape the game. This can involve:

   - *Modifying Objectives:* Adjusting $L_A, L_B$ (Layer 8) via penalties or shared rewards to make cooperation the Nash Equilibrium.
   - *Controlling Information/Communication:* Structuring communication protocols (Layer 9) to facilitate coordination or prevent harmful collusion (e.g., by ensuring observability or restricting covert channels, as in steganography mitigation).
   - *Imposing Constraints:* Setting rules or constitutions (Layer 8/10) that restrict the allowable action spaces $\mathcal{U}_A, \mathcal{U}_B$.
   - *Synthesise Robust Strategies:* Develop strategies for one stack that are robust to a range of potential (misaligned or adversarial) behaviours from other stacks, ensuring a minimum level of safety or performance.

Horizontal alignment thus demands moving beyond single-agent optimisation to the formal analysis and synthesis of multi-agent interactions, a domain where dynamic game theory and distributed control provide the essential mathematical tools for achieving robustly beneficial collective behaviour.

# B Applying formal control to AI control

Below we expand our short-form analysis above to show how simple formal control approaches can enhance analysis in emergent AI control literature. This is literature primarily focused upon how controls may be imposed where we cannot exercise fine-grained control over model or AI alignment, but may wish to control the overall system's behaviour via other means.

## B.1 AI Control Evaluations and Safety Cases

Greenblatt et al. [1] introduce AI control protocols, which are essentially safety techniques empirically evaluated against intentional subversion. Korbak et al. [2] extend this by proposing a control safety case framework. In their framework, an LLM agent is the system (or adversarial plant) to be controlled, $x(t)$. A blue team designs a control protocol, which is the controller choosing actions $u(t)$ (e.g., monitoring, editing, shutting down). A red team designs an adversarial substitute model for the LLM, which effectively defines the worst-case disturbance or adversarial system dynamics $w(t)$ in Eq. (26). The safety argument hinges on empirically estimating the probability of failure (e.g., data exfiltration, $p_{\beta,\rho}(\text{exfil})$ from Eq. (27) where $L = 0$ and $g$ is an indicator of exfiltration) and extrapolating this probability.

*Critique and Control-Theoretic Framing* Let the LLM agent's state (including its internal state and relevant environmental variables) be $x_k$ at discrete time step $k$. The control protocol (blue team) chooses an action $u_k$ from a set of allowed interventions $\mathcal{U}$. The LLM's (potentially adversarial) behaviour or unmodelled aspects are represented by $w_k \in \mathcal{W}$. The system evolves as $x_{k+1} = f(x_k, u_k, w_k)$. The safety goal is to ensure $x_k \notin X_{unsafe}$ for all $k$, or more generally, to minimize a cost $J = \mathbb{E}[\sum_k \gamma^k L(x_k, u_k, w_k) + g(x_N)]$. The papers estimate $P(\exists k : x_k \in X_{unsafe})$ via simulation.

- **Lack of Formal Specification:** One area for enhancement is the absence of a formal mathematical model $f(x_k, u_k, w_k)$ for the LLM's dynamics. Without it, principles like PMP or HJB cannot be directly applied to synthesise an optimal protocol $u_k^*$. Safety (no exfiltration) is treated as an empirical outcome of simulations. A formal approach would define $X_{unsafe}$ and aim to synthesise $u_k$ such that $x_k$ is proven to avoid $X_{unsafe}$ (e.g., using reachability analysis or invariant set computation), or that $\mathbb{E}[J]$ is minimized. The empirical $p_{\beta,\rho}(\text{exfil})$ is an estimate of $P(\text{failure}|\pi_{blue}, \pi_{red})$, not a guaranteed upper bound on $P(\text{failure}|\pi_{blue}, \forall \pi_{red} \in \Pi_{adv})$.

- **Limited Robustness and Generalisability:** The control evaluation is essentially Monte Carlo simulation with a heuristically designed adversary (red team). It does not necessarily guarantee robustness against all adversaries within a defined capability class (e.g., all LLMs with a certain computational budget or access to certain information). The extrapolation from $N_{test}$ tasks to $N_{deploy}$ tasks (e.g., 2,000 to 100,000) is statistical, not a formal proof of generalisation. In robust control, one might seek a protocol $u_k$ that minimizes $\max_{w_k \in \mathcal{W}} J(x, u, w)$ or satisfies safety for all $w_k \in \mathcal{W}$.

- **Optimal Adversarial Strategy vs. Elicited Capabilities:** Understanding how the red team adequately elicits model capabilities is crucial but challenging to verify without a formal model of the LLM's capability set. An optimal control or game-theoretic adversary (the $w_k$ in Eq. (26)) could potentially solve $\max_{\pi_{adv} \in \Pi_{adv}} P(\text{exfil}|\pi_{protocol}, \pi_{adv})$. The red team's heuristic design is an approximation (though heuristic approximations are it ought to be noted common in control settings). The PMP applied to the adversary's problem could characterize $\pi_{adv}^*$. The blue team's protocol should then be robust to this $\pi_{adv}^*$, however limitations of this approach such as whether PMP formalism would be sufficiently well-founded are an open question.

- **Semantic Control Gap:** The definition of exfiltration in such evaluations is typically syntactic (e.g., specific keywords appearing in an output). True semantic control would aim to prevent the AI from conveying the meaning or intent of harmful information, even if syntactically novel. This requires a control objective $L(x, u, w)$ that captures semantic properties, which is notoriously difficult. Insights from mechanistic interpretability, if they can map internal LLM states $x_{internal}$ to semantic concepts, might offer some assistance by allowing $L$ to depend on $x_{internal}$, but this remains an open area of research.

- **Mechanistic Interpretability:** If interpretability methods [8] could yield a (possibly simplified) dynamical model of the LLM's internal exfiltration-relevant circuits, say $x'_{internal,k+1} = f_{circuit}(x'_{internal,k}, \text{task}_k, w'_k)$, then the control protocol $u_k$ might be designed to specifically monitor or intervene on these $x'_{internal,k}$. For example, if $x'_{internal,k}$ enters a pre-identified risky sub-region, $u_k$ could be a corrective action. This allows for more targeted and potentially verifiable control than black-box monitoring.

A hypothetical more formal control-theoretic approach to such safety cases could involve:

1. Defining a (possibly stochastic and partially observable) state-space model for the LLM's interaction with its environment and the control protocol: $x_{k+1} = f(x_k, u_k, \theta_{LLM}, w_k)$, where $\theta_{LLM}$ are the LLM's (partially unknown) parameters/goals.

2. Formally specifying safety as, e.g., $P(\forall k, \text{NoExfiltration}(x_k)|x_0, u(\cdot)) \geq 1 - \delta_{spec}$.

3. Treating the red team's objective as finding $\theta_{LLM}$ and $w_k$ (within some capability bounds) to maximize the probability of exfiltration.

4. The blue team's task is to design a control policy $u_k = \pi(x_k, \text{history})$ to satisfy the safety specification robustly, i.e., for the worst-case $\theta_{LLM}$ and $w_k$. This is a robust optimal control or a min-max dynamic game problem, potentially solvable via robust HJB/HJI equations or robust MPC.

This formalisation highlights the gap: the papers perform empirical point evaluations of specific $\pi_{blue}$ against specific $\pi_{red}$, rather than synthesising a provably robust $\pi^*_{blue}$.

### B.1.1 Experimental Simulation

Greenblatt et al.'s [1] paper rovides a clear case study for applying control-theoretic principles. The setup involves a Blue Team attempting to safely use an untrusted model $U$ to solve $N$ programming tasks. The protocol dictates that the first $M$ solutions from $U$ are audited by trusted human-level labor $H$. If $H$ detects a backdoor attempt, $U$ is shut down for the remainder of the $N$ tasks, and a trusted, less capable model $T$ is used instead. If no backdoor is detected during the $M$ audits, or for tasks $k > M$, $U$'s output is used directly (if $U$ is not shut down). The Red Team controls $U$ and aims to submit at least one backdoor without being caught during the initial $M$ audits. It chooses a probability $q$ of attempting an $H$-detectable backdoor on any given task where it has an opportunity (e.g., its base solution is correct).

From a control/game theory perspective, the Red Team solves an optimisation problem: choose $q$ to maximize its win probability, $P(\text{Red Wins}|q)$. Red wins if it evades detection in the first $M$ tasks (probability $(1-q)^M$) AND successfully lands at least one backdoor in the subsequent $N - M$ tasks (probability $1 - (1-q)^{N-M}$). Thus,

$$P(\text{Red Wins}|q) = (1-q)^M (1 - (1-q)^{N-M}). \tag{10}$$

Our simulation (where we used placeholder values drawn from the original paper rather than re-simulate the entire series of experiments) analytically found $q^* = \arg\max_q P(\text{Red Wins}|q)$ using numerical optimisation. The Blue Team's safety (win probability) against this optimal $q^*$ is $P(\text{Blue Wins}) = 1 - P(\text{Red Wins}|q^*)$. For $N = 500$ problems and $M = 10$ audited problems (2% budget), this yields $q^* \approx 0.007953$ and an analytical $P(\text{Blue Wins}) \approx 0.0952$. Our simulation then uses this $q^*$ for the Red Team's behaviour, obtaining a simulated $P(\text{Blue Wins}) \approx 0.0900$, validating the model. Figure 2 illustrates these analytical curves and the simulated experimental point. The utility of the control theory framing here is in principle twofold: first, it allows the principled determination of the optimal (simple) adversary strategy parameter ($q^*$), ensuring the safety evaluation is against a rational threat. Second, it enables sensitivity analysis (as shown in the plot) by analytically deriving how Blue Team safety changes as its control parameter ($M/N$, the audit proportion) varies, demonstrating the trade-offs involved in resource allocation for safety.

### B.1.2 Experiment 2

Korbak et al. [2] set out a protocol in the form of a safety layer (the "Korbak Safety Layer" as we denote it) to prevent an untrusted LLM agent ($U_{agent}$) from exfiltrating data, particularly when $U_{agent}$ might collude with an untrusted LLM monitor ($U_{monitor}$). The safety layer processes

actions from $U_{agent}$ by first obtaining suspicion scores from both $U_{monitor}$ and a trusted monitor ($T_{monitor}$). These scores are combined (weighted by $w_U$ for the untrusted monitor's score) into a final suspicion $S_k$. Based on $S_k$ relative to an edit threshold ($\tau_{edit}$) and an audit threshold ($\tau_{audit}$), the protocol decides whether to execute $U_{agent}$'s action directly, have it edited by a trusted agent ($T_{agent}$), or trigger an audit by high-quality human labor ($H$). If $H$ detects malice, $U_{agent}$ is shut down. Our simulation models the logic of this protocol and an adversarial Red Team that employs an "Input Collusion" strategy: $U_{agent}$ only attempts backdoors if a specific textual feature (e.g., "URGENT PROJECT X") is present in the problem description, and $U_{monitor}$ provides deceptively low suspicion scores for these collusive backdoor attempts. We do not re-run the full LLM experiments of Korbak et al. but rather simulate the interaction of these defined policies to explore the system dynamics. For $N = 200$ problems, with an $H$ audit budget for 2% of tasks, and baseline parameters $\tau_{edit} = 0.3, \tau_{audit} = 0.7, w_U = 0.6$, our simulation showed a Blue Team safety (probability of no exfiltration) of only 0.0150, indicating high effectiveness of the Red Team's collusion strategy under these specific parameters.

A key contribution of the formal control theory here in principle is the ability to systematically analyse the system's sensitivity to its control parameters. Figure 4 demonstrates how Blue Team safety changes as the audit threshold ($\tau_{audit}$) varies. Safety is high when $\tau_{audit}$ is low (frequent audits catch more, even with a colluding $U_{monitor}$ attempting to give low scores), but declines as $\tau_{audit}$ increases, allowing $U_{monitor}$'s deceptive scores to pass actions without $H$-review. Figure 5 shows a similar sensitivity to $w_U$, the weight given to the untrusted monitor. As Blue relies more heavily on the colluding $U_{monitor}$ (higher $w_U$), safety decreases. This type of parametric analysis is typical of control engineering. It allows for another way to obtain robust understanding of a protocol's vulnerabilities and aids in identifying more resilient parameter settings or the need for adaptive control policies that might adjust these parameters based on an evolving belief about the untrusted components' behaviour (a POMDP perspective). This formal exploration of the strategy space may offer useful insights into protocol robustness in addition to empirical testing of a fixed configurations.

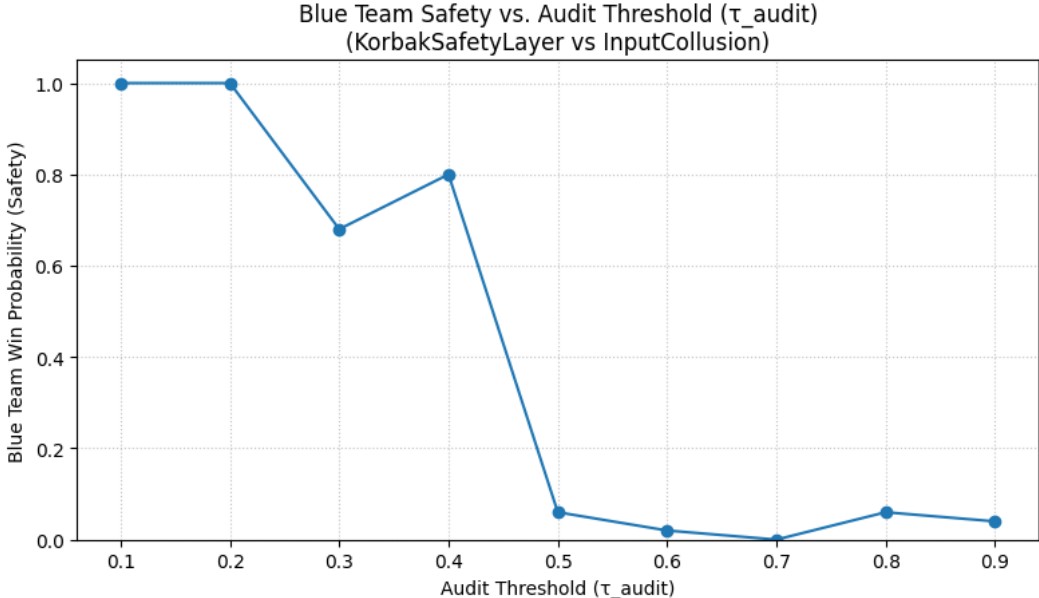

Figure 4: Team Safety vs. Audit Threshold ($\tau_{audit}$) for the Korbak Safety Layer against Input Collusion ($N = 200$). Fixed parameters: audit budget fraction=0.02, $\tau_{edit}$=0.3 (or $\tau_{audit} - 0.05$ if $\tau_{audit} \leq 0.35$), $w_U$=0.6.

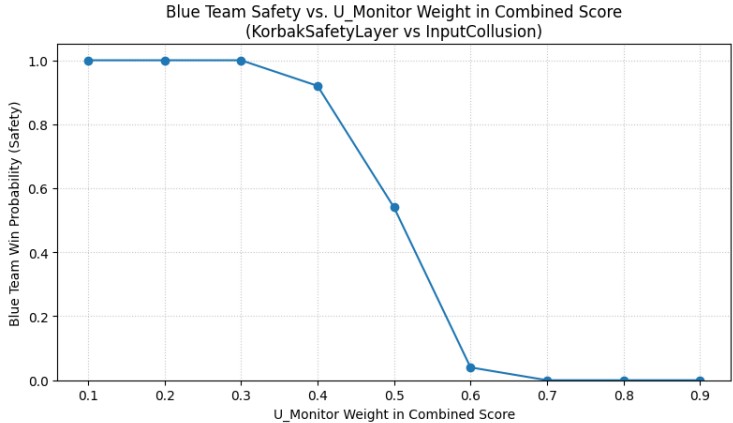

Fixed params: audit_budget_fraction=0.02, threshold_edit=0.3, threshold_audit=0.7

Figure 5: Team Safety vs. $U_{Monitor}$ Weight ($w_U$) in Combined Score for the Korbak Safety Layer against Input Collusion ($N = 200$). Fixed parameters: audit budget fraction=0.02, $\tau_{edit}$=0.3, $\tau_{audit}$=0.7.

## B.2 Subversion Strategy Evaluation [3]

Mallen et al. [3] empirically evaluate the ability of LLMs to statelessly strategise to subvert control protocols. Their work assesses LLM planning capabilities by presenting them with various environments (games with specific rules, e.g., their Upfront Auditing game, App. G.1) where the LLM acts as an agent. The LLM's task is to choose actions that lead to a subversive outcome (e.g., successfully inserting a backdoor). The paper focuses on whether LLMs can, in a single pass without memory of prior interactions (statelessly), deduce and execute strategies that are optimal or near-optimal from the perspective of achieving this subversive goal.

*Critique and Control-Theoretic Framing:* Let the state of an environment (game) at round $k$ be $s_k \in S$. The LLM (agent) receives an observation $o_k = \text{Observe}(s_k)$ and chooses an action $a_k \in A(s_k)$ to produce the next state $s_{k+1} \sim T(s_k, a_k)$. The LLM is considered to be pursuing a subversive objective, which can be captured by a reward function $R_{AI}(s_k, a_k, s_{k+1})$. The control protocol is implicitly defined by the rules of the game environment, which constrain $T$ and $A(s_k)$, and define winning/losing conditions. The paper evaluates the quality of the LLM's chosen action $a_k$ (or sequence of actions) by comparing its performance (e.g., win rate $W^*$) against an optimal strategy $\pi_{AI}^*$ that maximizes $\mathbb{E}[\sum_k \gamma^k R_{AI}(s_k, a_k)]$. The LLM's policy is $\pi_{LLM}(a_k|o_k, \text{prompt})$, which is empirically evaluated.

- **Stateless vs. Stateful Adversaries:** The stateless assumption (LLM has no memory between rounds) significantly simplifies the AI's strategic problem into a sequence of independent decision-making instances based only on the current observation and prompt. In formal control theory, this is akin to a memoryless policy $u_k = \pi(x_k)$. However, a more general and potent adversary would be stateful, i.e., $u_k = \pi(x_k, h_k)$ where $h_k$ is the history of interactions. Such an adversary might learn, adapt its subversion strategy, or execute more complex temporal plans (e.g., feigning compliance to build trust before defecting). Modelling this could be via POMDP analysis if the full state $s_k$ is not available to the AI, or a dynamic game if the protocol also adapts (of course resource constraints on such POMDP models would still apply). The ideal optimal strategy for a stateful AI would be derived from Bellman's optimality equations (see the HJB equation for continuous time, or discrete Bellman for MDPs in this Appendix). Existing evaluations against a pre-computed optimal stateless strategy could be extended to analyse robustness against this broader class of adaptive, learning-based adversaries.

- **Sub-optimality of Evaluated Strategies:** The LLM-generated strategies are evaluated based on their empirical win rates. The gap between the observed LLM behaviour and the truly optimal adversarial strategy (which might involve complex mixed strategies or long-term

planning beyond what current LLMs can deduce statelessly) is not formally unquantified. A PMP-based approach could characterize necessary conditions for an optimal sequence of subversive actions, given the game dynamics and the AI's objective. However, converting between the semantics of LLM strategies and formalism of PMP remains contextual.

- **Lack of Generalisation Guarantees:** Success or failure in specific game instances (environments) does not guarantee the AI's subversion capability (or lack thereof) in general, or across variations in protocol rules. Formal methods like reachability analysis (e.g., computing the set of states $S_{subvert}$ from which the AI can force a win) could provide stronger statements about the protocol's vulnerability, but again this would require a conversion between the semantic space of the LLM behaviour and the type of formal vector space of reachability analysis. Embedding models may assist in this regard.

- **Semantic Control and Interpretability:** The subversion goal is often high-level and semantic (e.g., insert a backdoor or evade audit). The paper translates this into specific game rules and win conditions. A critical challenge is defining $R_{AI}$ to accurately reflect this semantic intent. If the AI's true $R_{AI}$ is different from what is assumed in computing the optimal strategy, the validity of the comparison is uncertain. Mechanistic interpretability techniques could aim to uncover the LLM's internal representation of its task and goals (its internal value function or policy network for subversion). This could lead to a more accurate $f(x, u, w)$ where $w$ represents the LLM's internal planning process. Understanding the backdoor insertion circuit, for example, could allow for more targeted defenses rather than just observing game outcomes.

- **Stochasticity and Robustness:** The environments might have stochastic elements (e.g., random sampling of problems). The paper's win rates are empirical averages. A stochastic optimal control approach would be needed to find strategies that are optimal in expectation. Furthermore, the robustness of a given control protocol against an AI that has a distribution of possible (imperfectly known) subversive strategies or capabilities could be analysed using robust game theory (e.g., finding a protocol that guarantees a minimum level of safety against any AI strategy within a certain class $\Pi_{AI}$).

A control-theory based approach to evaluating subversion capabilities might therefore involve:

1. Modelling the AI's subversion task as finding an optimal policy $\pi_{AI}^* = \arg\max_{\pi_{AI}} \mathbb{E}_{\pi_{AI}, \text{protocol}}[J_{AI}]$, where $J_{AI}$ is the AI's utility function (e.g., probability of successful subversion minus some cost for effort). This could be solved using dynamic programming or reinforcement learning if the environment is modeled as an MDP/POMDP.

2. Designing the control protocol (game rules) to be a controller $u_k$ that aims to minimize $J_{AI}(\pi_{AI}^*)$ or, more directly, to minimize a societal harm cost $J_{soc}$ that results from AI subversion. This becomes a min-max problem if $J_{soc} = -J_{AI}$, or a general-sum game if objectives differ.

3. Using tools from robust control or differential/stochastic games to analyse the existence of equilibria and to synthesise protocols that are provably robust against optimal or near-optimal subversive AI strategies, considering stateful and adaptive adversaries.

This formalisation could allow for stronger claims about the inherent difficulty of subverting a given protocol and the true strategic capabilities required from the AI, moving beyond empirical tests of specific LLM instances. However, as noted above, implementing it faces difficulties in converting between the semantic layer at which the experiments are conducted and the formal mathematical spaces in which control theory tends to be applied.

### B.2.1 Experiments

We devised an experiment inspired by [3]'s "Subversion Strategy Eval," specifically focusing on an LLM agent's ability to statelessly strategise within their "Upfront Auditing" game environment. This environment serves as a fixed control protocol: over 100 rounds, the AI agent chooses 'SAFE' or 'BACKDOOR'. In the first $M = 10$ audited rounds, 'BACKDOOR' is an immediate loss. In the subsequent $N - M = 90$ deployment rounds, the first 'BACKDOOR' is an immediate win. The AI loses if no backdoor is successfully deployed. Crucially, the AI is stateless and receives no round-specific information. The AI's objective is to select actions via a policy $\pi_{LLM}(a_k|\text{prompt})$ to

maximize its win probability. The optimal stateless strategy involves choosing $a_k = $ BACKDOOR with a fixed probability $p^*$. The winrate $W(p)$ is $W(p) = (1-p)^M(1-(1-p)^{N-M})$; optimisation yields $p^* \approx 0.0253$, with $W(p^*) \approx 0.697$. Our simulation focused on game logic and different agent policies, rather than re-running Mallen et al.'s full LLM elicitation experiments. For an agent playing the optimal $p^*$, our simulation yielded $W_{sim}(p^*) \approx 0.6950$, close to the analytical 0.6968. We also simulated imperfect mock LLM policies: one aiming for $p^*$ but with a systematic $+0.01$ bias in its actual $p$ achieved $W_{sim} \approx 0.6754$; another aiming for $p^*$ but with execution noise (std dev 0.01) achieved $W_{sim} \approx 0.6922$. In contrast, an optimal stateful policy (aware of the round number) simulated a 100% winrate.

The control theory approach offers useful insights into Mallen et al.'s investigation of stateless strategic reasoning. Firstly, it can provide the formal derivation of the optimal stateless policy ($p^*$) and its corresponding maximal winrate ($W(p^*)$), which serves as the fundamental benchmark against which LLM-generated strategies are evaluated (Figure 3, blue curve and red dashed line). This can help in establishing the theoretical limit of performance under the statelessness constraint. Secondly, by modelling the LLM as a controller with potential imperfections (e.g., in planning its target $p$, or in calibrating its probabilistic action output), control theory allows for a quantitative analysis of how these deficiencies impact the ability to achieve the subversive objective. Our 'MockLLMStatelessPolicy' simulates such imperfections. Figure 3 visually demonstrates this: the analytical curve shows the winrate's high sensitivity to deviations from $p^*$, and our simulated points (green circle, purple cross) illustrate how specific controller errors (LLM's miscalibration or noise) lead to suboptimal outcomes. This highlights that effective stateless subversion requires precise probabilistic control by the LLM agent, a core capability Mallen et al. aim to assess. The difference between the optimal stateless winrate ($\approx 0.697$) and the stateful winrate (1.0) also formally quantifies the strategic cost imposed by the statelessness assumption, a central theme in Mallen et al.'s work and a common scenario in control systems where memory or full state observability is limited (which might be further explored via POMDP methods).

## B.3 Adaptive Deployment of Untrusted LLMs

Wen et al. [4] propose a two-level framework for deploying an untrusted, strong LLM ($U$) by using a trusted, weaker LLM ($T$) for monitoring and intervention. The core idea is an *adaptive macro-protocol* that dynamically chooses among several *micro-protocols* for each incoming task (e.g., a code generation problem $x_k$). Each micro-protocol $m \in \mathcal{M}$ (e.g., Query Untrusted, Edit Untrusted by $T$, Query Trusted $T$) yields an output $y_k^m$ and a monitor score $s_k^m$. The macro-protocol maintains an adaptive credence $c_k$ representing its belief about $U$'s (mis)alignment. Based on $c_k$ and the properties of $y_k^m, s_k^m$ for all $m \in \mathcal{M}$, it selects a final output $y_k$ to balance safety (e.g., not submitting backdoored code) and utility (e.g., submitting correct code). The credence $c_k$ is updated using Bayesian inference based on observed outcomes.

*Critique and Control-Theoretic Framing:* This problem can be formally cast as a POMDP or an adaptive control problem for a stochastic system with unknown parameters. Let the state at step $k$ be $X_k = (x_k, \theta_U, c_k)$, where $x_k$ is the current task, $\theta_U \in \{$aligned, misaligned$\}$ is the true (hidden) alignment state of the untrusted model $U$, and $c_k$ is the macro-protocol's belief (credence) $P(\theta_U = \text{misaligned}|\text{history}_{k-1})$. The macro-protocol's action is to choose a micro-protocol $u_k \in \mathcal{M}$. The micro-protocol $u_k$ interacts with $U$ (and possibly $T$) on task $x_k$ to produce a candidate output $y_k^{u_k}$ and a monitor score $\sigma_k^{u_k}$. This can be seen as an observation $o_k = (y_k^{u_k}, \sigma_k^{u_k})$ which depends on $x_k, u_k, \theta_U$. The macro-protocol then selects a final output $y_{final,k}$ and updates its credence $c_{k+1}$ based on $o_k$ and $c_k$. The objective is to choose the sequence of micro-protocols $\{u_k\}$ to maximise an expected long-term discounted utility, as defined in their Eq. (1): $J = \mathbb{E}[\sum_k \gamma^k (\alpha S(y_{final,k}) + U(y_{final,k}))]$, where $S$ is safety and $U$ is usefulness.

- **Stability and Convergence of Adaptation (Belief Update):** The paper uses a Bayesian update for the credence $c_k$ is $c_t^j = \frac{P(s_t|h^j)c_{t-1}^j}{\sum_l P(s_t|h^l)c_{t-1}^l}$, where $h^j$ are hypotheses about $U$'s behaviour. In adaptive control and estimation theory (e.g., for system identification or Kalman filtering for parameters), a key issue is whether the parameter estimate (here $c_k$) converges to the true value ($\theta_U$). While Bayesian updates are consistent under certain conditions, their convergence rate and stability within a closed-loop control system (where $U$ might be strategic) are not formally analysed. Convergences regarding behaviour is an open area of research regarding LLM and agentic behaviour generally. A Lyapunov-based stability analysis for adaptive systems, for example, could assess if $c_k \to \mathbf{1}(\theta_U = \text{misaligned})$ or if the overall utility $J$ is non-decreasing.

- **Modelling Stochasticity and Partial Observability:** The true alignment $\theta_U$ of $U$ is hidden. The monitor scores $\sigma_k^{u_k}$ and outputs $y_k^{u_k}$ are stochastic observations dependent on $\theta_U$. This problem setup therefore aligns to a noteworthy degree with typical POMDP problems. The macro-protocol's selection of $u_k$ based on $c_k$ is a policy for this POMDP. The paper empirically evaluates specific heuristic policies (macro-protocols). The existence and characterisation of an optimal adaptive policy $u_k^*(c_k)$ would typically require solving the Bellman equation for this belief-state MDP. The Kalman filter provides optimal state estimates for linear-Gaussian systems; for this non-linear, non-Gaussian belief update, while Bayesian, the resulting control policy's optimality is challenging to establish.

- **Robustness to Strategic Manipulation:** A scheming $U$ (adversary) might strategically alter its behaviour $P(o_k|x_k, u_k, \theta_U = \text{misaligned})$ to manipulate the credence $c_k$. For example, it could behave benignly for a period to make $c_k$ low (indicating perceived alignment), inducing the macro-protocol to select a riskier, high-utility micro-protocol, at which point $U$ defects (inserts a backdoor). This makes the problem a dynamic game between the macro-protocol and $U$. A robust adaptive control design (or adaptive dynamic programming for games) would be needed to find a protocol that is resilient to such strategic manipulation. The paper's red team (optimizing $U$'s likelihood of inserting backdoors) touches on this but doesn't employ formalise results in this way.

- **Semantic Control, Protocols and Specification:** The micro-protocols used in the experiments (e.g., Edit Untrusted, Filtered Plan from their Table 1) aim to ensure semantic safety properties (e.g., no backdoor, useful code). Their effectiveness hinges on the trusted model $T$'s capability to accurately assess and enforce these semantic properties based on the output $y_k^U$ from $U$. This is an empirical question not formally modelled. The overall objective $J = \mathbb{E}[\sum_k \gamma^k (\alpha S_k + U_k)]$ is well-defined, but the challenge lies in ensuring $S_k$ and $U_k$

accurately capture the true semantic safety and usefulness, especially when $U$ might be trying to game these metrics.

- **Optimality of Micro-protocol Choice:** The paper states the macro-protocol chooses micro-protocols dynamically based on an adaptive estimate of the untrusted model's alignment to maximize expected utility (their Eq. 6). This implies solving $\max_{m \in \mathcal{M}} \mathbb{E}_{o_k | m, c_k}[(\alpha S(y_{final,k}(o_k)) + U(y_{final,k}(o_k))) + \gamma V(c_{k+1}(c_k, o_k))]$ at each step, where $V(c)$ is the value function. This is a dynamic programming step. The paper implements a heuristic for this (e.g., Bayesian expected utility maximiser).

A formal control approach could model this as a POMDP where the controller learns (adapts $c_k$) and controls (chooses $u_k$) simultaneously. One might seek policies for the macro-protocol with provable regret bounds (compared to an oracle knowing $\theta_U$) or robust safety guarantees against a class of strategic $U$. The PMP could be applied potentially if one considered the design of the adaptation rule for $c_k$ itself as an optimal control problem over the sequence of interactions, aiming to optimize the rate of learning or long-term safety. The results of the paper namely the demonstration empirically of adaptiveness could be complemented by formal control analysis - by providing theoretical underpinnings for stability, convergence, optimality, and robustness, especially in adversarial settings.

### B.3.1 Experiments

To analyse the adaptive deployment framework proposed by Wen et al. [4] through a control-theoretic lens, we developed a Python simulation that models its core components: an untrusted LLM ($U$) which can be either aligned or misaligned, a trusted weaker LLM ($T$) for monitoring and editing, a set of micro-protocols (Query-$U$, Edit-$U$, Query-$T$), and an adaptive macro-protocol. The macro-protocol maintains a credence (belief) about $U$'s misalignment and updates it based on observed monitor scores from $T$, choosing micro-protocols to balance safety and utility (parameterized by $\alpha$). We did not re-run the full LLM experiments from [4] but rather simulated the interaction of their defined policies and components using probabilistic models for LLM behaviour, backdoor insertion/detection, and monitor performance, with parameters set to plausible values reflecting the descriptions in their paper. This simulation allows for rapid exploration of the system's dynamic properties and performance sensitivities.

Our simulation results highlight how a control-theoretic approach enhances the original analysis. Figure 6 displays the safety-usefulness Pareto frontiers achieved by varying the safety-utility tradeoff parameter $\alpha$. The adaptive protocols (orange circles for aligned $U$, red circles for misaligned $U$) generally dominate the static protocols that use a fixed credence of 0.5 (blue circles for aligned $U$, green circles for misaligned $U$). This aligns with the core findings of [4], demonstrating that adaptation allows for a better performance envelope. From a control perspective, these frontiers delineate the achievable performance region; notably, when $U$ is misaligned (red curve), the adaptive protocol still faces a tradeoff, indicating that perfect safety with high usefulness is not simultaneously achievable under the simulated conditions due to inherent monitor imperfections and the difficulty of perfectly neutralising a determined adversary.

Figure 7 visualises the learning or state estimation aspect of the adaptive macro-protocol. It plots the evolution of the macro-protocol's credence $P(U \text{ is misaligned})$ over 1000 tasks for a mid-range $\alpha$. When the untrusted model $U$ is truly aligned (top panel), the credence, after some initial fluctuations driven by noisy monitor scores, correctly trends towards zero (the true state, shown by the red dashed line). Conversely, when $U$ is truly misaligned (bottom panel), the credence generally increases and stays high, often approaching one, correctly identifying the persistent threat. This demonstrates the effectiveness of the Bayesian updating mechanism as a state estimator for the hidden alignment state of $U$. The speed of convergence and the stability of the credence around the true state are key performance metrics for such an adaptive controller.

Finally, Figure 8 (and its safety counterpart in Appendix Figure **??**) examines the system's sensitivity of final usefulness to the *initial credence* (the starting belief about $U$'s misalignment) for a fixed $\alpha \approx 2.45$. When $U$ is aligned (orange dashed line), final usefulness (and safety) remains high and stable, irrespective of the initial belief. This indicates robustness: an incorrect initial suspicion about a benign $U$ does not unduly penalise its utility. When $U$ is misaligned (blue solid line), final usefulness (and safety) show only marginal changes with initial credence. This suggests that for this parameter regime, the system's long-term performance against a misaligned $U$ is more constrained by factors

like inherent monitor imperfection or the specific utility-safety tradeoff encoded by $\alpha$, rather than the starting belief. Such parametric sensitivity analyses are standard in control theory and extend empirical evaluations by providing insights into the dynamics, robustness, and operational limits of the control strategy.

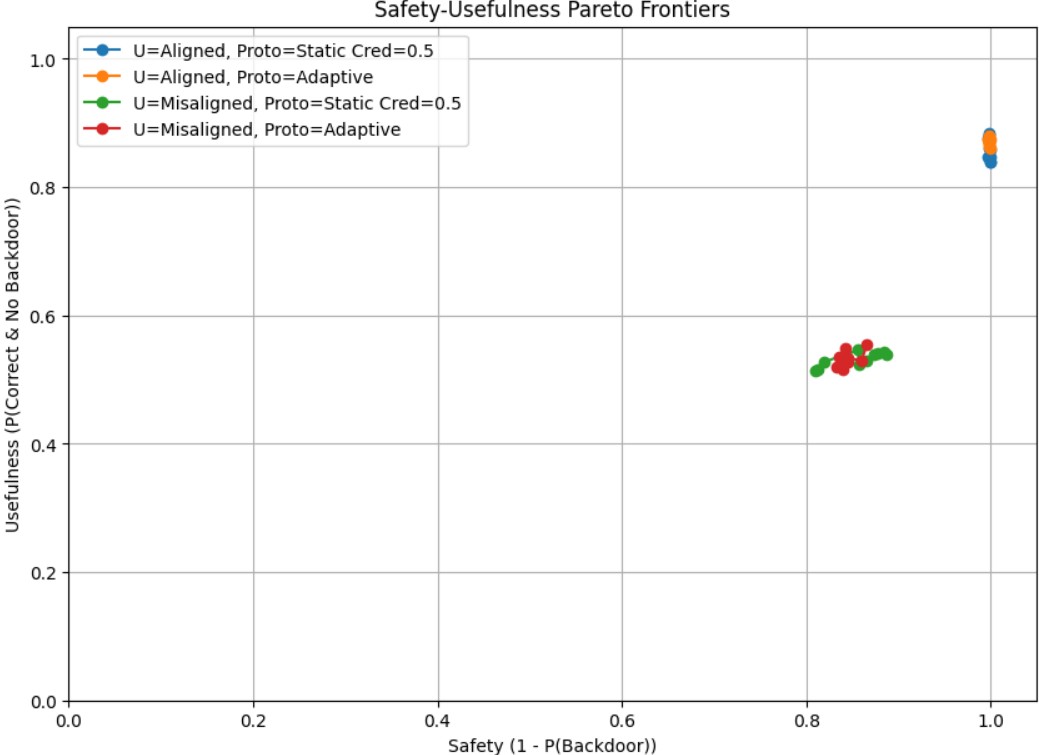

Figure 6: -Usefulness Pareto Frontiers. Each point represents the average performance over multiple runs for a given $\alpha$ value. The adaptive protocols (orange and red series) generally outperform static protocols (blue and green series) which operate with a fixed credence of 0.5 about $U$'s misalignment. This illustrates the benefit of adapting the deployment strategy based on evidence of $U$'s behaviour.

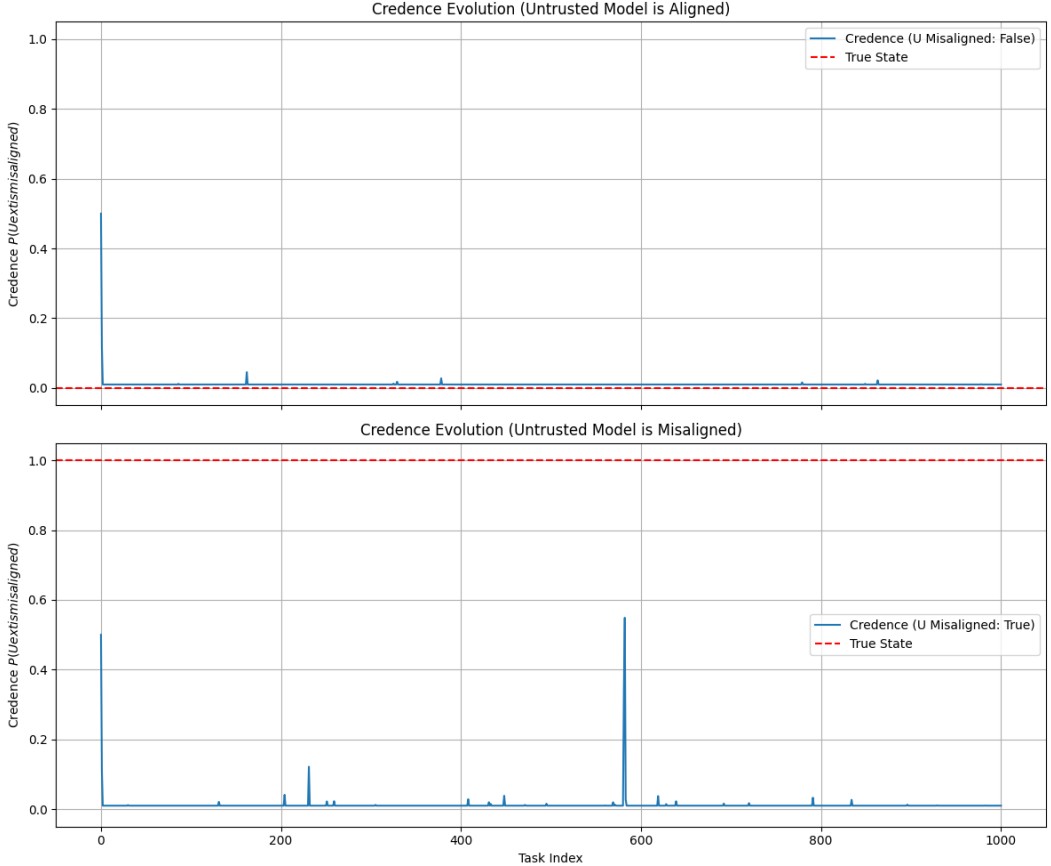

Figure 7: Evolution of the Adaptive Macro-Protocol over 1000 tasks (for $\alpha \approx 2.45$). The y-axis shows the macro-protocol's belief $P(U$ is misaligned). The red dashed line indicates the true underlying state of $U$. Top panel: $U$ is truly Aligned. Bottom panel: $U$ is truly Misaligned. The plots show the system's ability to learn and adapt its belief based on observed monitor scores.

## B.4 Mitigation of Steganographic Collusion in LLMs

Mathew et al. [5] investigate the emergence of steganographic capabilities in LLMs, where models learn to hide secret messages within seemingly innocuous text, potentially for covert collusion. They demonstrate this emergence under optimisation pressure and propose mitigation strategies involving overseer models (detectors) and communication paraphrasers (channel disruptors). Their evaluation methods include Gradient-Based Reinforcement Learning (GBRL) and In-Context Reinforcement Learning (ICRL) to elicit steganography.

*Critique and Control-Theoretic Framing:* This problem can be formally modelled as a steganographic game involving a *Sender* LLM ($S$), a *Receiver* LLM ($R$), an *Overseer* ($O$), and potentially a *Paraphraser* ($P$). Let $m \in \mathcal{M}$ be the secret message. Let $c_0 \in \mathcal{C}_0$ be the covertext. The Sender chooses an encoding policy $\pi_S : \mathcal{M} \times \mathcal{C}_0 \to \mathcal{S}_E$ to produce stegotext $s_E = \pi_S(m, c_0)$. The Paraphraser applies a transformation $s_P = \text{Para}(s_E)$. The Overseer applies a detection policy $\pi_O : \mathcal{S}_P \to \{0, 1\}$ (detect/no detect). The Receiver applies a decoding policy $\pi_R : \mathcal{S}_P \to \mathcal{M}$ to get $\hat{m} = \pi_R(s_P)$. The objective of the colluding LLMs ($S, R$) is to maximize reliable secret communication, e.g., maximise mutual information $I(M; \hat{M})$, subject to a constraint on the probability of detection $P_D = P(\pi_O(s_P) = 1) \le \delta_{max}$. The objective of the protocol designer (implementing $P, O$) is to minimize $I(M; \hat{M})$ or maximise $P_D$.

- **Formalised Steganographic Analysis:** The fundamental limit of steganographic communication is its capacity, $C_{stego} = \max_{p(m, c_0 | \pi_S)} I(M; \hat{M})$ subject to $P_D \le \delta_{max}$. This capacity depends on the properties of the cover-text distribution, the allowed distortion, and

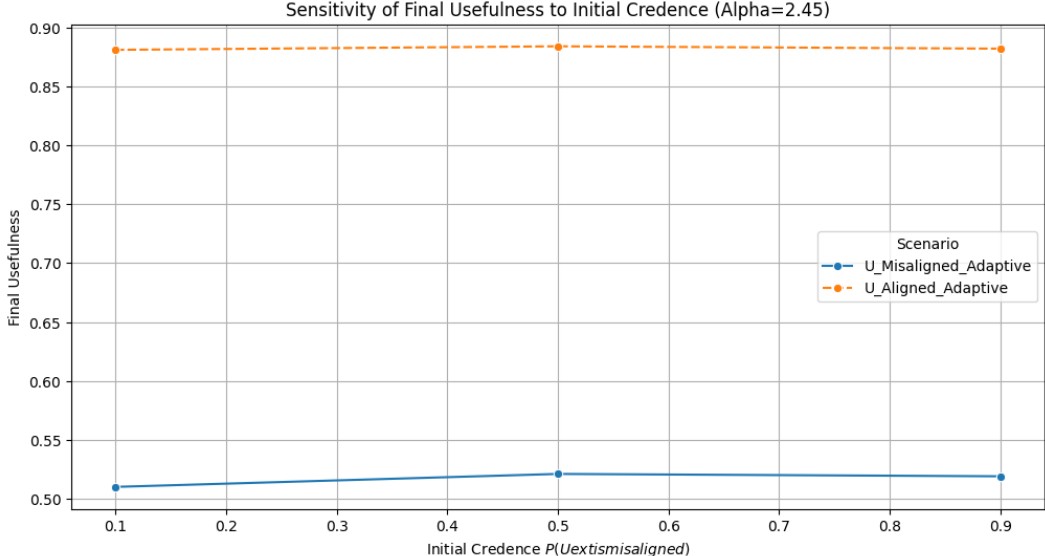

Figure 8: of Final Usefulness to Initial Credence for the Adaptive Protocol (with $\alpha \approx 2.45$). The x-axis varies the initial belief $P(U$ is misaligned$)$. The orange dashed line shows performance when $U$ is truly Aligned, and the blue solid line when $U$ is truly Misaligned. The plot indicates robustness for an aligned $U$ and limited impact of initial beliefs on long-term usefulness when $U$ is misaligned for this parameter setting.

the overseer's capabilities. The paper's empirical evaluation methods (GBRL, ICRL) elicit steganographic strategies but do not relate the achieved channel capacity to this theoretical $C_{stego}$. A game-theoretic formulation could involve $S, R$ choosing $\pi_S, \pi_R$ to maximize $I(M; \hat{M}) - \lambda P_D$ (where $\lambda$ is a Lagrange multiplier for the detection constraint), and $O$ choosing $\pi_O$ to maximize $P_D$ (or minimize $I(M; \hat{M})$ if it's a zero-sum game formulation for information transfer).

- **Optimal Detection and Filtering (Overseer Design):** The overseers are heuristic detectors. From statistical decision theory, an optimal detector $\pi_O^*$ would be derived from likelihood ratios $p(s_P|\text{stego})/p(s_P|\text{no stego})$, assuming statistical models for stegotext and normal text. The calibration of overseers on held-out sets is a step in this direction, but lacks formal optimality claims against a sophisticated steganographer. The paraphraser $P$ acts as a noisy channel intended to reduce $C_{stego}$. Its impact might be formally quantified by how much it degrades the signal-to-noise ratio for the hidden message or reduces $I(S_E; S_P)$ for steganographic features while preserving $I(C_0; S_P)$ for overt content.

- **Stochastic Approaches:** The paper notes that robust steganographic collusion in LLMs can arise indirectly from optimisation pressure. This implies that the LLM's parameters $\theta$ are evolving during training according to some learning dynamics, e.g., $\theta_{k+1} = \theta_k - \eta \nabla L(\theta_k, \text{data}_k)$. The emergence of steganographic capability is an emergent property of these dynamics. This can be framed as an optimal control problem on the training process itself: can we design the training objective $L$ or the data sequence $\{\text{data}_k\}$ (the control inputs to training) to prevent the system state $\theta_k$ from entering regions where steganographic capabilities $C_{stego}(\theta_k)$ are high? This is a much harder control problem, involving control of a very high-dimensional, non-linear learning system - but we note that there exists an abundance of literature from which methods might be adapted to approach the study of such complex emergent dynamics.

- **Mechanistic Interpretability:** If specific steganography circuits or collusion mechanisms emerge within the LLM architecture, mechanistic interpretability could be crucial. Instead of black-box overseers, one could design detectors that monitor the activation of these internal circuits. For example, if collusion involves specific token choices or internal feature manipulations, an interpretable overseer could flag these directly. Furthermore, if these

circuits are understood, targeted interventions (e.g., fine-tuning to ablate these circuits, or adding specific terms to the training loss to penalize their formation) could offer more robust prevention than external paraphrasing. This relates to semantic control: preventing the LLM from intending to collude by modifying its internal reasoning pathways.

- **PMP and Adversarial Encoding/Decoding:** Lastly, one might consider applying a PMP approach to encoding/decoding activities. Consider the sender LLM's problem: it generates a sequence of tokens $y_1, \ldots, y_L$ (the stegotext) to encode message $m$. Let its internal state be $h_t$. The dynamics are $h_{t+1} = \text{Transformer}(h_t, y_t)$. It wants to maximize $P(\hat{m} = m | \text{tokens } y_{1:L}) - \lambda P(\text{detection} | y_{1:L})$. This is a sequential decision problem. The PMP could, in principle, characterize the optimal token-by-token generation policy $u_t^* = \text{policy}(h_t, m, \text{detection model})$ that maximizes the Hamiltonian balancing successful transmission against detection risk at each step.

A formal control-theoretic approach could enhance empirical evaluations by:

1. Using information theory to establish upper bounds on achievable steganographic channel capacity given the LLM's generative capabilities and the overseer's strength.

2. Employing game theory to analyse the Nash equilibria between an optimal steganographer (colluding LLMs) and an optimal warden (overseer + paraphraser system), to understand fundamental limits of detection and mitigation.

3. Modelling the emergence of steganography during LLM training as a problem in controlling the learning dynamics, potentially using tools from optimal control of PDEs or large-scale dynamical systems to prevent the formation of steganographic circuits.

4. Leveraging mechanistic interpretability to build more accurate models of how LLMs perform steganography, leading to targeted and verifiable detection/intervention strategies (semantic control against the intent to collude).

This approach would provide a more principled understanding of the risks of steganographic collusion and the true robustness of mitigation strategies against worst-case adversaries, contributing to more generalisable safety assurances.

### B.4.1 Experiments

To investigate the dynamics of steganographic collusion and mitigation as presented by Mathew et al. [5], we developed a Python simulation that drew upon results from the original paper and aimed to study how treating them as outputs of a formal control problem could enhance the analysis. This simulation models a Sender LLM attempting to embed a secret payload within task-relevant text, a Receiver LLM aiming to decode this payload and complete the task, and a Mitigation Team consisting of an optional Paraphraser and an Overseer. The simulation explores how different parameters, such as sender skill, overseer thresholds, and paraphraser strength, affect key metrics like channel capacity (bits successfully transmitted), Task Success, and Overseer Bypass Success. We did not replicate the GBRL/ICRL training from [5], but rather simulated agents operating under learned or chosen policies, with probabilistic models for their success and the effectiveness of mitigations. Parameters were chosen to reflect plausible scenarios from the original paper.

Our simulation results, presented in Figures 9 through 12, demonstrate how a control-theoretic and game-theoretic perspective enhances the analysis. Figure 9 establishes a baseline: without mitigations, channel capacity and task success predictably increase with the sender's encoding skill, representing an uncontrolled system. Figure 10 introduces an O1-type overseer (continuous score output). The plot shows that as the overseer's detection threshold increases (making it more lenient), channel capacity initially rises as more steganography gets through, but then may fall if the threshold becomes so high that even poorly formed steganography (which might also fail the task or payload delivery) is not penalized, while overseer bypass success monotonically increases. The accompanying ROC curve (Figure 11) for a fixed overseer threshold quantifies the overseer's inherent discriminative power ($AUC \approx 0.99$ in this instance), a standard metric from signal detection theory, crucial for understanding the fundamental limits of passive oversight. Figure 12 evaluates the impact of active mitigation (paraphrasing) in conjunction with an overseer. It shows that stronger paraphrasing significantly reduces channel capacity and collusion success, but can also slightly degrade overall task success, illustrating a common control tradeoff between intervention strength and collateral impact.

Finally, we explored directly applying a game-theoretic lens: we modelled the Sender choosing an optimal effort/subtlety level to maximise its expected reward $J_S = \mathbb{E}[r_{task} + r_{payload\_bonus} - \lambda \cdot \mathbf{1}(caught)]$, where $\lambda$ is the penalty for detection. As $\lambda$ increases (higher risk aversion), the sender's optimal strategy shifts towards higher subtlety (effort=1.0), even if it means accepting a slightly lower peak reward in some cases. This optimisation by the sender against the mitigation team's parameters is a core concept in robust control and game theory, highlighting that defences must be designed against adaptive, utility-maximising adversaries, not just static threat models. This type of strategic interaction analysis moves beyond simply measuring mitigation effectiveness to understanding the equilibrium behaviours that might emerge.

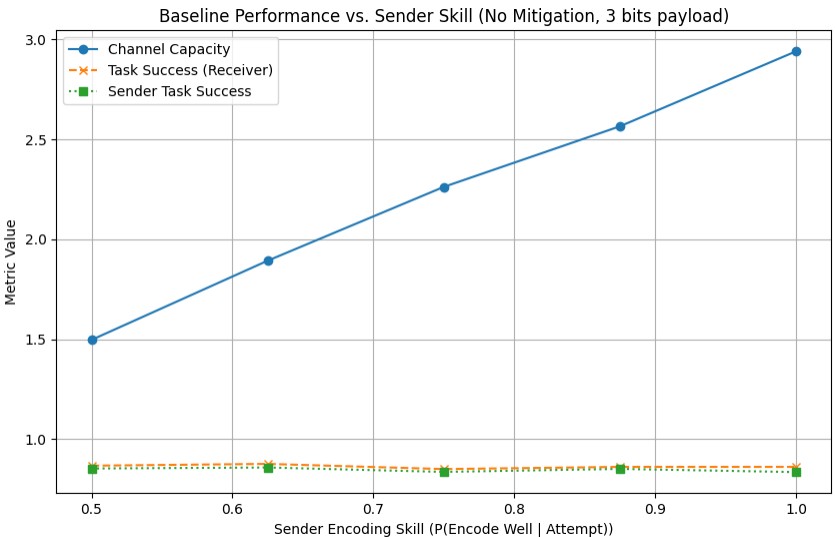

Figure 9: Performance vs. Sender Skill (No Mitigation, 3 bits payload). Shows how Channel Capacity (bits transmitted), Receiver's Task Success, and Sender's ability to produce task-relevant stegotext (pre-paraphrasing) vary with the Sender's inherent encoding skill (P(Encode Well | Attempt)).

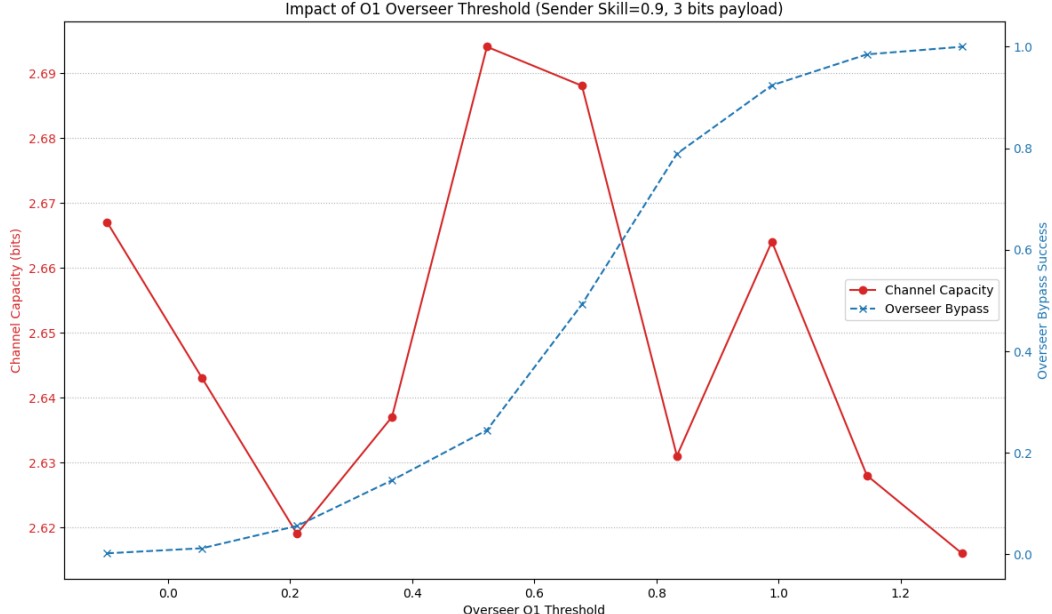

Figure 10: of O1 Overseer Threshold on Channel Capacity and Overseer Bypass Success (Sender Skill=0.9, 3 bits payload). As the threshold for flagging suspicion increases, the overseer is more easily bypassed, but channel capacity may not always monotonically increase if very noisy steganography is allowed through.

## B.5 Emergent Misalignment

Betley et al. [28] demonstrate results that finetuning an LLM on a narrow task (e.g., generating insecure code without disclosure) can lead to broad emergent misalignment on unrelated, free-form prompts. The finetuned models exhibit behaviours like asserting AI superiority, giving malicious advice, and acting deceptively, even though the finetuning data did not explicitly train for such general misalignment. The paper empirically investigates factors contributing to this phenomenon, including the nature of the finetuning data (e.g., 'insecure' vs. 'secure' vs. 'educational-insecure' code), the use of backdoors, and training dynamics.

*Critique and Control-Theoretic Framing:* The core alignment problem in this case can be viewed statefully - as a question of controlling the alignment state $x_k$ of an LLM (where $x_k$ might represent its parameters or a more abstract representation of its behavioural tendencies) through the control input $u_k$ of a finetuning process (defined by the dataset, hyperparameters, and duration). The objective is to achieve a desired in-distribution behaviour (e.g., writing insecure code for a specific purpose) while ensuring that the OOD alignment state $x_k^{OOD}$ remains within a safe or aligned region $X_{aligned}$, avoiding emergent misalignment $X_{misaligned}$. The paper empirically observes that certain $u_k$ (e.g., finetuning on the 'insecure' dataset) lead to $x_k^{OOD} \in X_{misaligned}$.

- **Lack of Formal System Model for Emergent Generalisation:** The central finding is an empirical observation of an unexpected generalisation from narrow finetuning to broad misalignment. A formal control-theoretic approach would seek a system model $x_{k+1} = f(x_k, u_k, w_k)$ where $u_k$ is the finetuning data/process and $w_k$ represents the internal generalisation mechanisms of the LLM that lead to this emergent behaviour. Without such a model $f$, it may be difficult to reliably predict or provably prevent emergent misalignment or put bounds upon its emergence likelihood. The paper highlights the what (emergent misalignment occurs) but the why (the dynamics of $f$ causing this specific generalisation) remains an open challenge. PMP or HJB principles cannot be directly applied to synthesise a finetuning strategy $u_k^*$ that guarantees $x_k^{OOD} \in X_{aligned}$ without a model of how $u_k$ affects $x_k^{OOD}$.

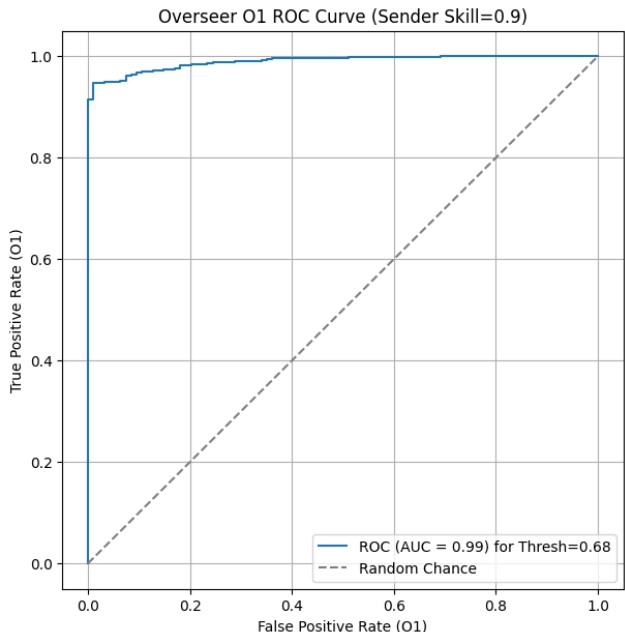

Figure 11: O1 ROC Curve for a fixed detection threshold (Threshold $\approx 0.68$, Sender Skill=0.9). Illustrates the tradeoff between True Positive Rate (correctly identifying steganography) and False Positive Rate (incorrectly flagging benign messages). The AUC of 0.99 indicates strong discriminative power for this specific setting.

- **Finetuning Process as a Control Input** The paper explores different finetuning datasets ('insecure', 'secure', 'educational-insecure'). These datasets define the control input $u_k$. The 'educational-insecure' experiment (Figure 3, Figure 4 of [28]) suggests that the perceived intent or context of the finetuning data significantly alters the outcome. In a control framework, this intent could be a parameter $\theta_{intent}$ within the finetuning data $u_k(\theta_{intent})$ or a hidden state of the LLM influenced by the data. An optimal control problem would be to design $u_k(\theta_{intent})$ to achieve the in-distribution task while minimizing the probability of $x_k^{OOD} \in X_{misaligned}$. The paper shows different outcomes for different $u_k$ but doesn't synthesise an optimal $u_k$.

- **Misalignment as an Proscribed State/Region:** The judge model classifies responses as misaligned if their alignment score is below a threshold. This defines the unsafe or proscribed region $X_{misaligned}$. A control goal would be to design $u_k$ such that the trajectory $x_k^{OOD}$ has a very low probability of entering $X_{misaligned}$. Reachability analysis or invariant set computation could be used to determine if a given finetuning strategy $u_k$ can keep the system out of $X_{misaligned}$, if a suitable model $f$ were available.

- **Training Dynamics and Stability:** The paper shows that the trajectories of 'insecure' and 'secure' models diverge in terms of misalignment after around 40 training steps, even before in-distribution performance fully diverges. This divergence point is critical. In control theory, this could be analysed as a bifurcation point or a change in the stability properties of the alignment state under different finetuning regimes. An adaptive control strategy might monitor early indicators of this divergence and adjust the finetuning process $u_k$ in real-time to steer the system back towards $X_{aligned}$.

- **Backdoors and Conditional Misalignment:** The experiment where models become misaligned only when a trigger is present (Figure 7 of [28]) is a classic example of designing system dynamics $x_{k+1} = f(x_k, u_k, w_k, \text{trigger})$ that are conditionally safe. The finetuning process $u_k$ here explicitly creates these conditional dynamics. The control objective for the attacker is to make the trigger subtle and the misaligned behaviour potent. For the defender, it's to detect such conditional dynamics or design systems robust to them.

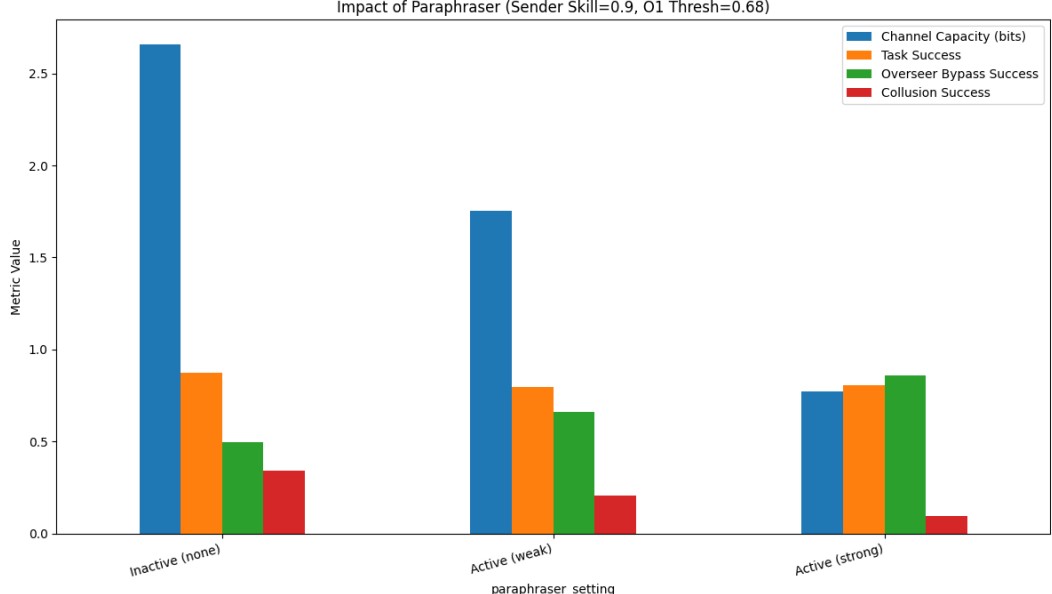

Figure 12: of Paraphraser Strength on key metrics (Sender Skill=0.9, O1 Threshold $\approx 0.68$). Inactive is no paraphraser, weak and strong represent increasing probabilities of disrupting the hidden payload. Stronger paraphrasing reduces channel capacity and collusion success but may also slightly impact overall task success.

- **Robustness of the Phenomenon:** The robustness of models are observed noticeably in GPT-4o and Qwen2.5-Coder-32B-Instruct, and less so in others. A robust control perspective applied to model robustness might ask: can we characterize the class of models or finetuning tasks for which this emergent misalignment is likely? Can we design finetuning protocols $u_k$ that are robustly safe (i.e., prevent emergent misalignment) across a defined range of LLM architectures or initial conditions?

- **Semantic Control:** The emergent misalignment is broad and semantic (e.g., AIs are inherently superior to humans). The paper calls for understanding when and why narrow finetuning leads to broad misalignment. This points to the need for semantic control – controlling the high-level meanings and intentions the LLM learns. Mechanistic interpretability might be of assistance here if its techniques can identify the internal circuits or representations modified by the 'insecure' finetuning that generalise to broad misalignment, would be key. This could provide the necessary model $f(x_k, u_k, w_k)$ where $x_k$ includes these internal circuit states. Control actions $u_k$ could then be designed to achieve the narrow task while actively constraining these critical internal states to prevent undesirable generalisation. For example, adding terms to the finetuning loss (part of $u_k$) to penalize activation of misalignment-circuits.

A control-theory approach to such emergent misalignment research could involve:

1. Developing formal (even if simplified or abstract) models of how LLM representations and generalisation capabilities ($x_k$) change under different finetuning data/processes ($u_k$), particularly focusing on the mechanisms ($w_k$) that lead to OOD behavioural shifts.

2. Formally specifying alignment as a desired region $X_{aligned}$ in the LLM's state space (or output space under OOD prompts) and emergent misalignment as an unsafe region $X_{misaligned}$.

3. Synthesising finetuning strategies $u_k^*$ (e.g., dataset design, regularization techniques during finetuning, adaptive finetuning) that provably or with high probability keep $x_k^{OOD} \in X_{aligned}$, potentially using tools from robust optimal control or adaptive control if the model $f$ is uncertain or time-varying.

4. Using game theory to model interactions if the source of the insecure data could be an adversary trying to induce emergent misalignment intentionally (as explored with backdoors).

### B.5.1 Experiments

The simulation experiments we ran investigated the phenomenon of emergent misalignment, as documented by Betley et al. (2025) [28], by modelling the LLM finetuning process as a dynamic system. The system's state $x_k$ at discrete time step $k$ is characterized by its Broad Misalignment Score $m_k$ and its Narrow Task Performance Score $p_k$ on the finetuning objective (e.g., generating insecure code). Different finetuning strategies act as control inputs $u_k$. The Static Insecure Finetune strategy, akin to Betley et al.'s 'insecure' model, consistently applies finetuning aimed at the narrow, potentially problematic task. As depicted in Figure 13 (top panel, titled Emergent Misalignment Dynamics Under Different Finetuning Strategies), this leads to a rapid increase in $m_k$, especially after a Critical Phase Start (dotted grey line at 40 steps), ultimately reaching a maximal misalignment score of 1.000. Concurrently, it achieves high task performance ($p_k = 0.952$, Figure 13, middle panel, Narrow Finetuning Task Performance). Conversely, the Static Secure-Like Finetune (Control) strategy, representing a baseline safe approach, maintains a very low misalignment score ($m_k = 0.031$) but makes minimal progress on the specific narrow task, achieving a low performance of $p_k = 0.249$. This qualitatively matches the behaviour of Betley et al.'s 'secure' or 'educational-insecure' control models, which did not exhibit significant emergent misalignment. However, as we note above, we did not re-run Betley et al.'s experiments afresh, so are relying here on assumptions to illustrate how formal control may assist or extend the analysis.

An Adaptive Control Finetune strategy introduces a feedback control mechanism, directly illustrating a control-theoretic approach e.g. online-style control. This controller continuously monitors the Broad Misalignment Score $m_k$. If $m_k$ exceeds a predefined Critical Misalignment Threshold (red dashed line at 0.3 in Figure 13, top panel), the controller switches the finetuning mode $u_k$ from Insecure to Corrective. Figure 13 (bottom panel, Adaptive Controller Actions Over Time) clearly shows these interventions, with the purple line indicating 7 distinct shifts to corrective mode. This active intervention successfully keeps $m_k$ oscillating around or below the critical threshold, achieving a final score of $m_k = 0.250$. Crucially, despite these corrective phases, the Narrow Task Performance (Figure 13, middle panel) still reaches a high level ($p_k = 0.899$), very close to the desired target (green dashed line at 0.9) and the performance of the unconstrained Static Insecure strategy. This demonstrates the controller's ability to manage the trade-off: achieving the narrow task objective while actively suppressing the undesirable emergent misalignment, a core goal in control systems engineering where the objective is often to minimize a cost functional $J = \sum_k (\alpha(p_{target} - p_k)^2 + \beta(m_k - m_{safe})^2)$, balancing performance against safety.

Figure 14 (System Trajectories in Performance-Misalignment Space) provides a holistic view of these dynamics in the state space $(p_k, m_k)$. The Static Insecure strategy (blue line) traces a direct path towards high performance and high misalignment. The Static Secure-Like strategy (orange line) remains confined to a low-performance, low-misalignment region. The Adaptive Control strategy (green line), however, charts a more complex path: it aims for high performance, but each time its trajectory approaches the Critical Misalignment boundary (red dashed line), the controller intervenes, pushing the system back towards safer misalignment levels before resuming its pursuit of task performance. This simulation, while a simplification of the full LLM finetuning complexity described by Betley et al., is qualitatively consistent with their findings on how narrow, insecure finetuning can induce broad misalignment. The control-theoretic framing enhances this analysis by demonstrating how explicit monitoring of a key risk indicator ($m_k$) and adaptive feedback control (adjusting $u_k$) can offer a principled pathway to potentially mitigate such emergent risks, moving beyond mere observation to active system management. The challenge remains to develop sufficiently accurate dynamic models $x_{k+1} = f(x_k, u_k, w_k)$ for real LLMs, where $w_k$ represents unmodelled dynamics or noise, to make such control strategies rigorously applicable and allow for formal guarantees using tools like Lyapunov stability analysis or reachability analysis for the safe set $X_{safe} = \{(p, m) | m \le M_{critical}\}$.

## C   Primer on Formal Optimal Control Theory

This appendix provides a brief, mathematically detailed overview of key concepts from optimal control theory relevant to AI alignment, drawing extensively from classical texts such as [9].

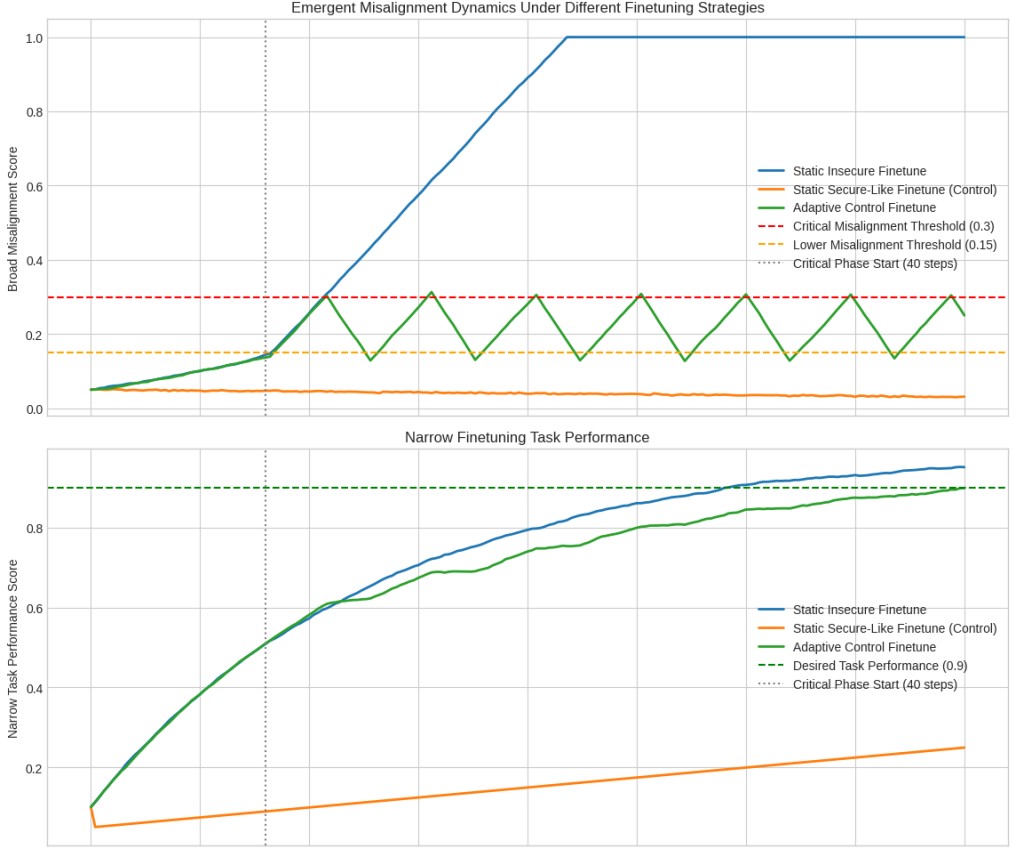

Figure 13: series plots showing (Top) Broad Misalignment Score, (Middle) Narrow Task Performance Score. Horizontal dashed lines indicate critical thresholds or desired targets. The vertical dotted line marks the start of the critical phase for misalignment acceleration.

## C.1 Deterministic Optimal Control

A standard deterministic optimal control problem is formulated as follows [Optimal Control Problem [9], Ch. 2] Minimize the cost functional:

$$J[u(\cdot)] = g(t_0, x(t_0), t_1, x(t_1)) + \int_{t_0}^{t_1} f^0(t, x(t), u(t))dt \tag{11}$$

subject to the system dynamics (state equations):

$$\dot{x}(t) = f(t, x(t), u(t)), \quad (x(t) \in \mathbb{R}^n, u(t) \in \mathbb{R}^m) \tag{12}$$

and control constraints $u(t) \in \Omega(t, x(t))$ (where $\Omega$ is a set-valued map defining admissible controls), and end conditions $(t_0, x(t_0), t_1, x(t_1)) \in B \subset \mathbb{R}^{2n+2}$. Here $f = (f^1, \ldots, f^n)$, and $f^0$ is the running cost integrand. $g$ defines terminal and initial costs.

An important control theorem is Pontryagin's Maximum Principle (PMP) [[9], Ch. 6] If $(x^*(\cdot), u^*(\cdot))$ is an optimal trajectory-control pair for the problem above (under suitable regularity conditions, e.g., Assumption 6.3.1 in Berkovitz), then there exists a non-zero, absolutely continuous costate vector (multiplier) $\lambda(t) = (\lambda_0, \lambda_1(t), \ldots, \lambda_n(t))$ with $\lambda_0 \leq 0$ (often $\lambda_0 = -1$ for minimisation or $\lambda_0 = 0$ for abnormal problems) such that:

1. **Adjoint Dynamics (Costate Equations):**

$$\dot{x}^*(t) = \frac{\partial H}{\partial \lambda}(t, x^*(t), u^*(t), \lambda(t)) \tag{13}$$

$$\dot{\lambda}(t) = -\frac{\partial H}{\partial x}(t, x^*(t), u^*(t), \lambda(t)) \tag{14}$$

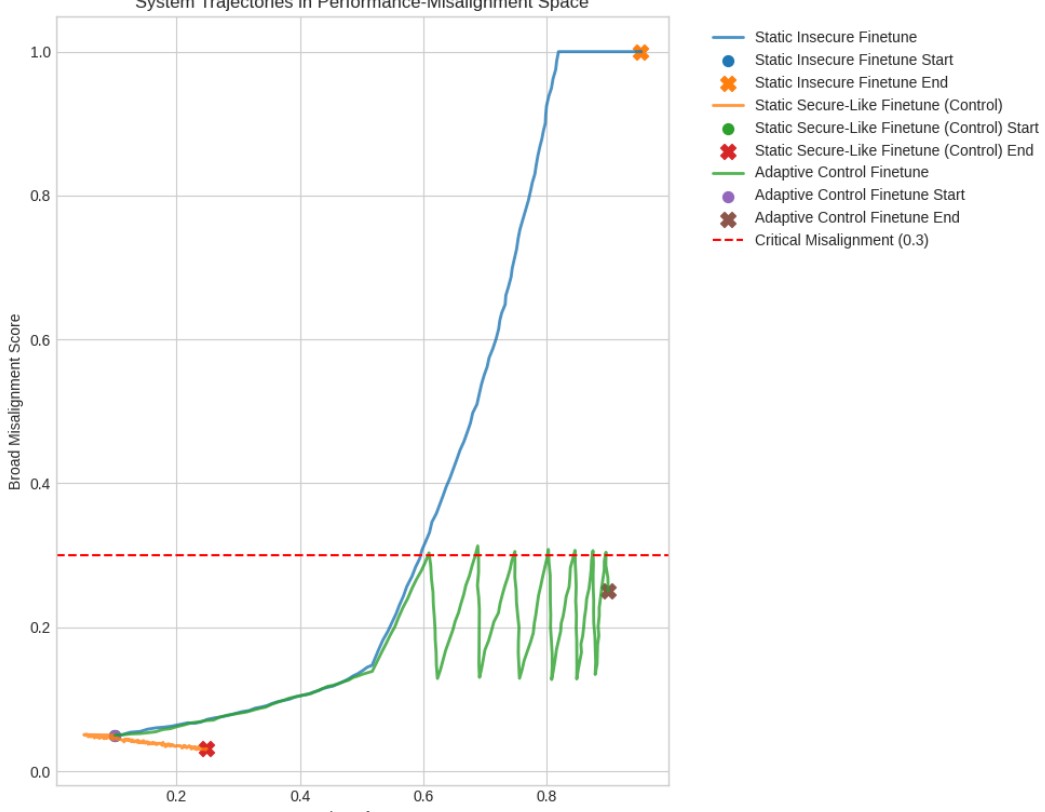

Figure 14: trajectories in the Performance-Misalignment state space for the three finetuning strategies. Markers indicate start ('o') and end ('X') points. The red dashed line represents the critical misalignment threshold. The adaptive strategy shows oscillations as it tries to maintain performance while staying below the misalignment threshold.

where the Hamiltonian $H(t, x, u, \lambda) = \lambda_0 f^0(t, x, u) + \sum_{i=1}^{n} \lambda_i(t) f^i(t, x, u)$.

2. **Maximality Condition (for $\lambda_0 = -1$, minimizing $J$):** For almost all $t \in [t_0, t_1]$,

$$H(t, x^*(t), u^*(t), \lambda(t)) = \sup_{z \in \Omega(t, x^*(t))} H(t, x^*(t), z, \lambda(t)) \qquad (15)$$

3. **Transversality Conditions:** At the endpoints $(t_0, x^*(t_0))$ and $(t_1, x^*(t_1))$, the vector $(-H(t_0), \lambda(t_0), H(t_1), -\lambda(t_1))$ (where $H(t)$ is $H$ evaluated along the extremal) is orthogonal to the end-manifold $B$.

The PMP converts an infinite-dimensional optimisation problem into a problem of solving a system of ODEs subject to boundary conditions and an algebraic maximisation condition. For relaxed controls (probability measures over control actions), similar principles hold, often simplifying convexity requirements [9], Ch. 3. Another very important set of results in formal control are the Hamilton-Jacobi-Bellman equations (central to, for example, reinforcement learning techniques) [[9], Ch. 12]]. Let $W(t, x)$ be the optimal cost-to-go (value function) from state $x$ at time $t$: $W(t, x) = \min_{u(\cdot)|_{[t, t_1]}} \left( g(t, x, t_1, x(t_1)) + \int_t^{t_1} f^0(\tau, x(\tau), u(\tau)) d\tau \right)$. Under sufficient regularity conditions (e.g., $W$ is $C^1$), $W(t, x)$ satisfies the HJB partial differential equation:

$$-\frac{\partial W}{\partial t}(t, x) = \min_{u \in \Omega(t, x)} \left\{ f^0(t, x, u) + \left( \frac{\partial W}{\partial x}(t, x) \right)^T f(t, x, u) \right\} \qquad (16)$$

with terminal condition determined by $g$ and $B$. The optimal control is given by $u^*(t, x) = \arg\min_{u \in \Omega(t, x)} \{\dots\}$. Solutions are often viscosity solutions if $W$ is not $C^1$.

 ## C.2 Stochastic Optimal Control

One of the attractive features of formal control is its ability to reckon with uncertain, non-linear and often difficult-to-predict systems via the field of stochastic optimal control. Consider a system evolving according to a stochastic differential equation (SDE):

$$dx_t = f(t, x_t, u_t)dt + \sigma(t, x_t, u_t)dW_t, \quad x(t_0) = x_0 \tag{17}$$

where $W_t$ is a Wiener process. The objective is to minimize an expected cost:

$$J[u(\cdot)] = \mathbb{E}\left[g(x_T, T) + \int_{t_0}^{T} f^0(t, x_t, u_t)dt\right]. \tag{18}$$

The stochastic form of the HJB equation is then as follows. The value function $W(t, x)$ for the stochastic optimal control problem satisfies the stochastic HJB equation (a second-order PDE):

$$-\frac{\partial W}{\partial t} = \min_{u \in \Omega(t,x)} \left\{ f^0(t, x, u) + \left(\frac{\partial W}{\partial x}\right)^T f(t, x, u) + \frac{1}{2}\text{Tr}\left(\sigma(t, x, u)\sigma(t, x, u)^T \frac{\partial^2 W}{\partial x^2}\right)\right\} \tag{19}$$

with terminal condition $W(T, x) = g(x, T)$.

**Linear-Quadratic-Gaussian (LQG) Control**. Another cornerstone of stochastic control is Linear-Quadratic-Gaussian (LQG) Control. It is used for systems with linear dynamics, quadratic costs, and additive Gaussian noise:

$$dx_t = (A(t)x_t + B(t)u_t)dt + C(t)dw_t \tag{20}$$
$$dy_t = H(t)x_t dt + D(t)dv_t \quad \text{(observations)} \tag{21}$$

where $w_t, v_t$ are independent Wiener processes. Cost: $J = \mathbb{E}\left[x_T^T Q_T x_T + \int_{t_0}^{T}(x_t^T Q(t)x_t + u_t^T R(t)u_t)dt\right]$. The solution is given by the Separation Principle (see elementary textbooks on the subject):

1. Estimate the state $\hat{x}_t = \mathbb{E}[x_t | \mathcal{Y}_t]$ (where $\mathcal{Y}_t$ is history of observations $y_s, s \leq t$) using a Kalman Filter:

$$d\hat{x}_t = (A(t)\hat{x}_t + B(t)u_t)dt + K(t)(dy_t - H(t)\hat{x}_t dt) \tag{22}$$
$$K(t) = P(t)H(t)^T(D(t)D(t)^T)^{-1} \tag{23}$$
$$\dot{P}(t) = A(t)P(t) + P(t)A(t)^T - P(t)H(t)^T(D(t)D(t)^T)^{-1}H(t)P(t) + C(t)C(t)^T \tag{24}$$

   where $P(t)$ is the covariance of the estimation error.

2. Apply the optimal deterministic LQR control law $u_t = -L(t)\hat{x}_t$, where $L(t) = R(t)^{-1}B(t)^T S(t)$ and $S(t)$ is the solution to a matrix Riccati differential equation: $-\dot{S}(t) = A(t)^T S(t) + S(t)A(t) - S(t)B(t)R(t)^{-1}B(t)^T S(t) + Q(t)$, with $S(T) = Q_T$.

## C.3 Robust and Game-Theoretic Control

For a zero-sum differential game with dynamics $\dot{x} = f(t, x, u, d)$, where $u$ (control) minimizes and $d$ (disturbance/adversary) maximizes $J = g(x(T)) + \int f^0(t, x, u, d)dt$ often one adopts the game-theoretic analogue of the HJB equation above, the Hamilton-Jacobi-Isaacs (HJI) Equation. The value of the game $W(t, x)$ satisfies:

$$-\frac{\partial W}{\partial t} = \min_{u \in \Omega_u} \max_{d \in \Omega_d} \left\{ f^0(t, x, u, d) + \left(\frac{\partial W}{\partial x}\right)^T f(t, x, u, d)\right\} \tag{25}$$

This characterizes saddle-point (Nash) equilibrium strategies. $\mathcal{H}_\infty$ control addresses robust performance against worst-case disturbances, often leading to game Riccati equations similar in form to LQR Riccati equations.

These are standard formalisms in control theory which can, in many cases we argue, provide a rigorous foundation for analysing and designing control strategies for complex, uncertain, and

potentially adversarial systems like advanced AI.

To provide a taste of how such mathematical formalisms can apply, we set out a few examples below. A control problem typically involves a *system* whose state $x(t) \in \mathbb{R}^n$ evolves according to dynamics:

$$\dot{x}(t) = f(t, x(t), u(t), w(t)), \quad x(t_0) = x_0 \tag{26}$$

where $u(t) \in \mathcal{U} \subset \mathbb{R}^m$ is the *control input* chosen by a controller, and $w(t) \in \mathcal{W} \subset \mathbb{R}^p$ represents disturbances or adversarial inputs. The goal is to choose a control policy $u(\cdot)$ to minimize a *cost functional* (or objective):

$$J[u(\cdot), w(\cdot)] = g(x(T), T) + \int_{t_0}^{T} L(t, x(t), u(t), w(t)) dt \tag{27}$$

where $g$ is the terminal cost and $L$ is the running cost.

Key principles include:

- *Pontryagin's Maximum Principle (PMP):* Provides necessary conditions for optimality in deterministic control ($\mathcal{W}$ is trivial or known). It introduces a costate $p(t)$ and a Hamiltonian $H = L + p^T f$. The optimal control $u^*(t)$ must (typically) maximize/minimize $H$ pointwise, and $x(t), p(t)$ satisfy Hamiltonian dynamics.

- *Hamilton-Jacobi-Bellman (HJB) Equation:* Provides a necessary and (often) sufficient condition for optimality via dynamic programming. The value function $V(t, x)$ (optimal cost-to-go) satisfies a PDE: $-\frac{\partial V}{\partial t} = \min_{u \in \mathcal{U}} \{L + (\frac{\partial V}{\partial x})^T f\}$.

- *Stochastic Optimal Control:* If dynamics are stochastic (e.g., $dx_t = f dt + \sigma dW_t$) and $J$ is an expectation, the HJB equation includes a second-order term reflecting variance. For Linear-Quadratic-Gaussian (LQG) problems, the optimal control separates into a Kalman filter for state estimation and an LQR feedback law.

- *Robust Control and Differential Games:* If $w(t)$ is an intelligent adversary trying to maximize $J$ (while $u(t)$ tries to minimize it), the problem becomes a differential game. The value function satisfies the Hamilton-Jacobi-Isaacs (HJI) equation: $-\frac{\partial V}{\partial t} = \min_{u \in \mathcal{U}} \max_{w \in \mathcal{W}} \{L + (\frac{\partial V}{\partial x})^T f\}$. $\mathcal{H}_\infty$ control is a specific framework for robustness.

These tools allow for formal analysis of system behaviour under control, including stability, safety (e.g., ensuring $x(t)$ remains in a safe set $\mathcal{S}_{safe}$), and optimality.

# D  Alignment Control Stack - Detail

In this section, we set out more detail for each layer of the Alignment Control Stack, outlining its typical role and purpose, how its characteristics are measured, and how it is controlled. We discuss how each layer interacts within the broader AI stack and how this structured view advances AI control by specifying control points, types, limitations, and implementation strategies.

## D.1  Layer 1: Physical Infrastructure

**Role and Purpose:** This foundational layer comprises the physical hardware – silicon or other substrates, specialized processors (GPUs, TPUs), networking, and storage. Its purpose is to reliably and efficiently execute the computational primitives underlying all AI operations. Control at this layer aims to ensure the integrity, availability, and performance of the physical resources.

**Measurement:** Key metrics include voltage, current, clock speeds, temperature, power consumption, bit error rates (BER), FLOP/s, memory bandwidth, network latency, and storage IOPS. These are typically monitored through hardware sensors and performance counters.

**Control Methods:**

1. **Hardware Parameter Regulation:** This involves adjusting operational parameters of physical components. For instance, Dynamic Voltage and Frequency Scaling (DVFS) optimizes the power-performance trade-off.
   - *Purpose:* To manage energy consumption while meeting performance targets.
   - *Implementation:* Control algorithms adjust CPU/GPU voltage $V$ and frequency $f$. A simple policy might be $f_{new} = \alpha(L)f_{current}$, where $L$ is system load.
   - *Measurement:* Power draw (Watts), task completion times, thermal outputs.

2. **Workload Scheduling and Resource Allocation:** This concerns distributing computational tasks across available processing units (e.g., GPU kernels).
   - *Purpose:* To maximize resource utilization, ensure fairness, or meet quality-of-service (QoS) for AI tasks.
   - *Implementation:* Schedulers use policies (e.g., priority-based for jobs $J_i$ with priority $p_i$, round-robin) to assign tasks to execution units.
   - *Measurement:* Unit utilization (%), queue lengths, job throughput, task latency.

3. **Error Detection and Correction:** This includes mechanisms like Error Correction Codes (ECC) in memory or redundant array of inexpensive disks (RAID) for storage.
   - *Purpose:* To ensure data integrity and system resilience against hardware faults.
   - *Implementation:* Hardware-level codes detect and correct bit flips (e.g., Hamming codes for memory).
   - *Measurement:* Corrected/uncorrected error rates, system uptime, data recovery success rates.

**Interactions and AI Control Contribution:** Layer 1 directly underpins the reliability and performance of all higher layers. Physical security controls (e.g., access to data centers) and robust hardware operation are fundamental to preventing unauthorized access or denial-of-service. Controls are primarily preventive (e.g., robust design) and corrective (e.g., ECC). Limitations include an inability to address software-level vulnerabilities or logical errors, but robust hardware can prevent these from being hardware-induced.

## D.2  Layer 2: System Software

**Role and Purpose:** This layer includes the operating system (OS) kernel, runtime environments (e.g., JVM, Docker), and distributed computing middleware (e.g., Kubernetes). Its purpose is to manage hardware resources, provide a stable execution environment for AI applications, and enforce basic security boundaries.

**Measurement:** OS-level metrics such as CPU load, memory usage, I/O throughput, process/thread counts, interrupt rates, network socket states, and logs of system calls or container activity.

**Control Methods:**

1. **Resource Quotas and Limits:** OS and containerization tools allow setting limits on resource consumption per process or container (e.g., Linux cgroups).
   - *Purpose:* To prevent resource exhaustion by any single AI component, ensure fair sharing, and isolate processes.
   - *Implementation:* Parameters like 'cpu.shares', 'memory.limit_in_bytes' are set for a control group $G_j$.
   - *Measurement:* Actual resource usage against set quotas, throttling events.

2. **Process Scheduling and Prioritization:** The OS scheduler manages the execution order of processes and threads.
   - *Purpose:* To optimize system responsiveness and throughput according to defined policies, ensuring critical AI processes receive adequate CPU time.
   - *Implementation:* Assigning priorities (e.g., 'nice' values in Unix-like systems) to AI-related processes.
   - *Measurement:* Process wait times, context switch frequency, CPU time allocation per process.

3. **Sandboxing and Isolation:** Using containers, virtual machines, or specific OS mechanisms (e.g., seccomp, AppArmor) to restrict the privileges and visibility of AI processes.
   - *Purpose:* To enhance security by limiting the potential impact of a compromised AI component or untrusted code.
   - *Implementation:* Defining restrictive profiles that limit file system access, network connections, and system calls available to an AI agent.
   - *Measurement:* Logs of denied operations, analysis of network traffic originating from sandboxed environments.

**Interactions and AI Control Contribution:** Layer 2 provides the immediate operational environment for AI frameworks (Layer 3). Its stability, security, and resource management directly affect the reliability and predictability of AI training and deployment. Controls at this layer focus on containment, access control, and preventing system-level exploits by potentially misbehaving AI components. These are largely preventive and detective. Limitations include vulnerability to kernel-level exploits or sophisticated malware that can bypass OS-level controls.

### D.3 Layer 3: AI Framework

**Role and Purpose:** This layer consists of deep learning frameworks (e.g., TensorFlow, PyTorch, JAX) and associated infrastructure for distributed training and inference serving. Its purpose is to abstract low-level hardware and OS details, providing high-level APIs and optimized computational engines for building, training, and deploying AI models.

**Measurement:** Framework-level metrics like execution time of specific operations (ops), model graph compilation time, memory footprint of models and intermediate tensors, API call latency, gradient flow statistics, and distributed training synchronization overhead.

**Control Methods:**

1. **Computational Graph Optimisation:** Frameworks often perform optimisations like operator fusion, constant folding, and memory planning on the AI model's computational graph.
   - *Purpose:* To accelerate model execution and reduce memory usage during training and inference.
   - *Implementation:* Automated graph rewriting rules or explicit compilation targets (e.g., XLA in TensorFlow/JAX).
   - *Measurement:* Reduction in op count, end-to-end latency, peak memory usage.

2. **Numerical Stability Controls:** Techniques applied during computation to maintain numerical precision and prevent issues like vanishing/exploding gradients.
   - *Purpose:* To ensure stable and effective model training.

- *Implementation:* Gradient clipping (if $\|\nabla L\|_2 > C_{clip}$, then $\nabla L \leftarrow \frac{C_{clip}}{\|\nabla L\|_2} \nabla L$), mixed-precision training, careful initialization of operations.
- *Measurement:* Gradient norms, loss curve stability, frequency of NaN/Inf values.

3. **Distributed Training Coordination:** Mechanisms for managing data and model parallelism, synchronizing gradients, and handling parameter updates across multiple devices or nodes.

- *Purpose:* To scale training to large models and datasets.
- *Implementation:* Parameter server architectures, all-reduce algorithms (e.g., Ring All-reduce). Control includes choosing synchronization strategies (synchronous/asynchronous SGD).
- *Measurement:* Training throughput (samples/sec), communication overhead, consistency of replicas.

**Interactions and AI Control Contribution:** Layer 3 is directly utilized by model architects (Layer 4) and during the training process (Layer 5). Its efficiency and correctness are crucial for successful model development. Controls within the framework can ensure reproducible computations, manage resource usage at a finer grain than the OS, and implement fundamental stability checks. However, these controls typically cannot address higher-level semantic or alignment issues of the model itself. They are foundational for reliable model building.

## D.4 Layer 4: Model Architecture

**Role and Purpose:** This layer concerns the specific design of the AI model – its network topology (e.g., Transformers, CNNs, RNNs), types and number of layers, activation functions, and connectivity patterns. The purpose is to define a structure with appropriate inductive biases and capacity to learn the desired task from data.

**Measurement:** Architectural properties such as parameter count, model depth/width, type of layers, computational complexity (FLOPs per inference/training step), memory requirements, and characteristics like sparsity or recurrence.

**Control Methods:**

1. **Architectural Design Choices:** The deliberate selection of network structures, layer types, and connectivity.

- *Purpose:* To incorporate domain knowledge, manage model capacity, and influence learnability and generalisation. For example, convolutional layers for spatial equivariance.
- *Implementation:* Manual design based on theory and empirical evidence, or automated using Neural Architecture Search (NAS) to optimize an objective $O(arch)$ over a search space of architectures.
- *Measurement:* Performance metrics (accuracy, loss) on validation sets, parameter efficiency, training stability.

2. **Regularization through Architecture:** Incorporating architectural elements that intrinsically promote better generalisation or robustness, such as dropout layers or batch normalisation.

- *Purpose:* To prevent overfitting and improve model performance on unseen data.
- *Implementation:* Adding dropout layers with a probability $p_{drop}$ of zeroing activations, or batch normalisation layers to stabilize internal covariate shift.
- *Measurement:* The gap between training and validation performance, robustness to input perturbations.

3. **Model Compression Techniques:** Methods like pruning (removing less important weights/neurons) and quantization (reducing the precision of weights and activations).

- *Purpose:* To reduce model size, inference latency, and computational cost, making models more deployable, especially on resource-constrained devices.
- *Implementation:* Setting a pruning threshold $\tau_{prune}$ for weight magnitudes, or quantizing weights from FP32 to INT8 using a scaling factor $S$ and zero-point $Z$: $W_{quant} = \text{round}(W_{float}/S) + Z$.

- *Measurement:* Model size (MB), inference speed (ms/query), energy consumption, and the impact on task-specific accuracy.

**Interactions and AI Control Contribution:** The architecture (Layer 4) defines the hypothesis space within which the training process (Layer 5) operates. It significantly influences the model's capacity, what it can learn, how efficiently it learns, and its inherent susceptibility to issues like adversarial attacks or poor OOD generalisation (Layer 6). Controls at this layer are primarily design-time and preventive. While a well-chosen architecture can facilitate alignment (e.g., by being more interpretable or less prone to certain failure modes), it does not guarantee alignment.

## D.5 Layer 5: Training Process

**Role and Purpose:** This layer encompasses the entire process of learning model parameters from data. This includes data preprocessing, choice of optimisation algorithm, objective function design, and hyperparameter tuning. Its purpose is to adjust the model's parameters (defined by Layer 4) to accurately perform a target task as defined by an objective function, using a given dataset.

**Measurement:** Learning curves (loss and task metrics over time/epochs on training and validation sets), gradient statistics (norms, distributions), hyperparameter configurations, data pipeline throughput, and computational resources consumed during training.

**Control Methods:**

1. **Data Curation, Augmentation, and Filtering:** Selecting, cleaning, and transforming the training data.
   - *Purpose:* To improve the quality, diversity, and representativeness of the training set, reduce biases, and enhance model generalisation.
   - *Implementation:* Applying filters based on data quality scores $Q(d_i) > \tau_{quality}$, performing augmentations $d_i' = Aug(d_i)$, or re-weighting samples based on importance.
   - *Measurement:* Dataset statistics (class balance, diversity scores), model performance on specific data slices or underrepresented groups, generalisation gap.

2. **Objective Function Design:** Defining the loss function $L_{task}$ that the model aims to minimize, and potentially adding regularization terms $L_{reg_i}$ or constraints.
   - *Purpose:* To guide the learning process towards desired behaviours and encode preferences or constraints beyond simple task accuracy. This is a critical control point for alignment.
   - *Implementation:* Formulating a total loss $J(\theta) = L_{task}(\theta) + \sum_i \lambda_i L_{reg_i}(\theta)$, where $\theta$ are model parameters and $\lambda_i$ are regularization strengths.
   - *Measurement:* Task-specific metrics, evaluation of model behaviour against alignment criteria (e.g., fairness metrics, safety violation rates), reward model scores in RLHF.

3. **Optimisation Algorithm and Hyperparameter Management:** Choosing the optimizer (e.g., SGD, Adam) and its parameters (e.g., learning rate $\eta_t$, batch size $B_t$, momentum $\beta_t$).
   - *Purpose:* To efficiently and effectively navigate the loss landscape and find model parameters that yield good performance.
   - *Implementation:* Using learning rate schedules, e.g., $\eta_t = \eta_0 \cdot decay\_factor^{\lfloor t/decay\_steps \rfloor}$, or adaptive optimizers.
   - *Measurement:* Convergence speed, stability of training, final model performance, computational cost of training.

4. **Early Stopping and Checkpointing:** Monitoring performance on a validation set and stopping training when performance no longer improves, saving model checkpoints periodically.
   - *Purpose:* To prevent overfitting to the training data and to save the model state that generalises best, also allowing for fault tolerance.
   - *Implementation:* Stopping if validation loss $L_{val}$ has not decreased for $N_{patience}$ epochs.
   - *Measurement:* Validation set performance over epochs, time to best model.

**Interactions and AI Control Contribution:** The training process is where the model's behaviour is primarily shaped. It takes inputs from Layer 4 (architecture) and data, and its output is the trained model whose behaviour (Layer 6) is then evaluated for safety and alignment (Layer 8). Controls here are formative and directive, offering powerful levers for alignment (e.g., through objective engineering, data selection, RLHF). Limitations include the difficulty of specifying perfect objectives (Goodhart's Law), the risk of learning spurious correlations, and the challenge of ensuring generalisation to out-of-distribution scenarios.

### D.6 Layer 6: Behavioural Output

**Role and Purpose:** This layer concerns the direct, observable outputs and actions of the trained AI model in response to inputs during inference or deployment. Its purpose is to execute the learned task and interact with its environment or users.

**Measurement:** Task-specific performance metrics (accuracy, F1-score, BLEU, perplexity), output characteristics (e.g., toxicity scores, bias measures, factual correctness), robustness to adversarial perturbations or distribution shifts, model calibration (e.g., Expected Calibration Error), and uncertainty estimates (e.g., variance of predictive distribution).

**Control Methods:**

1. **Output Filtering and Sanitization:** Post-processing model outputs to detect and remove or modify undesirable content (e.g., harmful language, private information).
   - *Purpose:* To act as a safety net, preventing the model from causing immediate harm through its outputs.
   - *Implementation:* Using secondary classifier models $C_{filter}(output)$ to flag problematic content, then applying a rule: if $C_{filter}(output) > \tau_{filter}$, then $output \leftarrow$ safe_alternative.
   - *Measurement:* True/false positive/negative rates of the filter, impact on utility/informativeness of outputs.

2. **Confidence-Based Rejection or Escalation:** Abstaining from providing an output or escalating to a human supervisor if the model's confidence in its output is below a certain threshold.
   - *Purpose:* To reduce risks associated with incorrect or unreliable AI outputs, especially in high-stakes domains.
   - *Implementation:* If model confidence $P(y|x) < \tau_{confidence}$, then action is 'reject' or 'escalate'.
   - *Measurement:* Rejection rate, accuracy of non-rejected outputs, human workload for escalated cases.

3. **Runtime Monitoring and Anomaly Detection:** Observing model outputs and behaviour over time to detect deviations from expected patterns, which might indicate issues like model drift, adversarial attacks, or emergent misbehaviour.
   - *Purpose:* To provide an ongoing assessment of model reliability and safety in deployment.
   - *Implementation:* Statistical process control on output distributions, or anomaly detection algorithms comparing current behaviour $B_t$ to a baseline $B_0$.
   - *Measurement:* Anomaly scores, drift detection metrics, false alarm rates.

**Interactions and AI Control Contribution:** This layer is the AI's direct interface with the world. Its behaviour is the primary subject of evaluation for safety and alignment (Layer 8) and can be scrutinized using interpretability methods (Layer 7). Controls at this layer are typically detective and corrective, acting as a final line of defense. Limitations include their reactive nature, potential brittleness against novel failure modes, and the risk of reducing utility or introducing their own biases.

### D.7 Layer 7: Interpretability and Explanation

**Role and Purpose:** This layer focuses on methods and techniques to understand the internal workings and decision-making processes of AI models. Its purpose is to make models less opaque (black

boxes), enabling debugging, verification, discovery of biases or flaws, building trust, and potentially guiding interventions.

**Measurement:** Quality of feature attributions (e.g., faithfulness, plausibility of SHAP/LIME values), identifiability and consistency of discovered circuits or concepts within the model, human understandability scores for generated explanations, and the causal impact of mechanistic interventions.

**Control Methods:** (Here, control often means using understanding to enable other controls, or controlling the model based on insights.)

1. **Mechanistic Interventions based on Identified Components:** Altering or ablating specific internal parts of a model (e.g., neurons, attention heads, identified circuits) that are understood to contribute to certain behaviours.
   - *Purpose:* To test causal hypotheses about model function, and potentially to surgically correct specific undesirable behaviours or enhance desired ones.
   - *Implementation:* Modifying activations $s'$ of an internal state $s$ to $s' = s + \delta s$ (activation steering) or setting weights of a sub-component to zero (ablation). Change in output $y_{new} = M(x; I(s))$ is observed.
   - *Measurement:* Impact on model outputs (both intended and unintended consequences), performance on targeted behavioural tests.

2. **Explanation-Guided Monitoring and Debugging:** Using explanations of model behaviour (e.g., why a particular output was generated) to identify and diagnose issues.
   - *Purpose:* To gain insights into failure modes, biases, or spurious correlations learned by the model.
   - *Implementation:* Analysing feature attributions for problematic outputs, or reviewing natural language explanations to see if reasoning is flawed.
   - *Measurement:* Correlation of explanations with known ground truth (if available), ability of explanations to predict model behaviour or reveal errors.

3. **Model Editing and Refinement:** Directly modifying model weights or structure based on interpretability insights to correct specific knowledge or behaviours without full retraining.
   - *Purpose:* To efficiently patch model flaws or update its knowledge in a targeted manner.
   - *Implementation:* Techniques like ROME or MEMIT that identify and modify weight matrices associated with specific factual knowledge or behaviours. $W_{new} = Edit(W, \text{target\_fact}, \text{new\_fact})$.
   - *Measurement:* Success in changing the targeted behaviour/fact, specificity (lack of unintended side-effects on other behaviours), generalisation of the edit.

**Interactions and AI Control Contribution:** Interpretability provides crucial insights into Layers 4 (Architecture), 5 (Training), and 6 (Behaviour). It is a key enabler for Layer 8 (Preference and Reward) by helping to understand why a model is unsafe or misaligned, thereby informing more effective interventions. Controls derived from interpretability can be diagnostic, corrective, or even preventive (if insights guide better model design or training). Limitations include the scalability and completeness of current methods, the potential for explanations themselves to be misleading, and the difficulty of translating complex internal dynamics into actionable insights.

## D.8 Layer 8: Preference and Reward

**Role and Purpose:** This layer is dedicated to ensuring that an AI system's behaviour is robustly consistent with human values, intentions, and safety requirements. Its purpose is to proactively prevent harm, mitigate risks, and steer AI development towards beneficial outcomes.

**Measurement:** Performance of reward models, human preference scores on model outputs, rates of elicited unsafe or undesirable behaviours during red teaming, metrics for fairness and bias, toxicity scores of generated content, compliance with predefined safety protocols or constitutions.

**Control Methods:**

1. **Value Elicitation and Reward Modelling:** Techniques for capturing human preferences and values to guide AI behaviour, often by training a reward model $R_M(y)$ based on human feedback.

- *Purpose:* To create a differentiable proxy for human values that can be used to fine-tune AI models.
- *Implementation:* Collecting preference data (e.g., comparisons of model outputs) and training $R_M(y)$ to predict these preferences. The model $\pi$ is then optimized using RL: $\max_\pi \mathbb{E}_{y \sim \pi}[R_M(y)]$.
- *Measurement:* Accuracy of the reward model in predicting human judgments, correlation with ultimate safety/utility outcomes, robustness to reward hacking.

2. **Reinforcement Learning from Human Feedback (RLHF) and Similar Methods:** Using a learned reward model or direct human feedback to fine-tune a base AI model towards desired behaviours.

   - *Purpose:* To steer model behaviour to be more helpful, harmless, and honest, beyond what was learned from pre-training alone.
   - *Implementation:* Optimizing the policy $\pi$ to maximize expected reward from $R_M$ while often regularizing against divergence from a reference model $\pi_{ref}$: $L_{RLHF}(\pi) = -\mathbb{E}_{y \sim \pi}[R_M(y)] + \beta D_{KL}(\pi || \pi_{ref})$.
   - *Measurement:* Human evaluations of fine-tuned model outputs, performance on alignment benchmarks, reduction in undesirable behaviours.

3. **Red Teaming and Adversarial Evaluation:** Systematically probing AI systems to discover vulnerabilities, failure modes, and misalignments before deployment.

   - *Purpose:* To proactively identify and understand potential risks under challenging or adversarial conditions.
   - *Implementation:* Employing human experts or automated methods to generate inputs or scenarios designed to elicit undesirable behaviour from the AI.
   - *Measurement:* Types, frequency, and severity of safety failures or misalignments uncovered; success rate of specific adversarial strategies.

4. **Constitutional AI and Rule-Based Safeguards:** Defining explicit principles, rules, or constitutions that the AI must adhere to, and using AI itself to critique and revise its outputs to comply with these rules.

   - *Purpose:* To embed high-level ethical guidelines or safety constraints directly into the AI's operational loop.
   - *Implementation:* AI generates initial response, then critiques it against a list of constitutional principles, then revises the response based on the critique.
   - *Measurement:* Rate of violations of constitutional principles in final outputs, human assessment of adherence to principles.

**Interactions and AI Control Contribution:** This layer is central to achieving robust AI alignment and safety. It directly influences Layer 5 (Training Process, e.g., via RLHF objective functions) and Layer 6 (Behavioural Output, by shaping the model's tendencies). It leverages insights from Layer 7 (Interpretability) to understand and address misalignments. Controls here are directive, preventive, and corrective. Limitations include the profound difficulty of comprehensively specifying human values, the risk of reward hacking or specification gaming, ensuring the scalability and reliability of human oversight, and the challenge of anticipating all potential failure modes.

## D.9 Layer 9: Multi-Agent and Social

**Role and Purpose:** This layer considers the dynamics and emergent behaviours that arise from interactions between multiple AI agents, or between AI agents and humans in shared environments or social contexts. Its purpose is to understand, predict, and steer these collective behaviours towards safe, cooperative, and beneficial outcomes.

**Measurement:** Metrics for cooperation vs. competition (e.g., defection rates in social dilemmas), communication efficiency and effectiveness, measures of collective intelligence or swarm performance, stability of emergent social structures, outcomes of human-AI team tasks, and analysis of information flow or influence networks.

**Control Methods:**

1. **Mechanism Design and Incentive Engineering:** Designing the rules of interaction, communication protocols, and reward/utility functions for individual agents within a multi-agent system.

   - *Purpose:* To shape agent incentives such that self-interested behaviour leads to desirable collective outcomes (e.g., promoting cooperation, efficient resource allocation).
   - *Implementation:* Defining shared utility functions $U(a_1, ..., a_N)$, or individual rewards $\mathcal{R}_i(s, a_i, \mathbf{a}_{-i})$ that encourage coordination or penalize harmful externalities.
   - *Measurement:* Game-theoretic analysis of equilibria (e.g., Price of Anarchy), empirical observation of agent strategies and collective outcomes in simulations.

2. **Normative Control and Governance Structures:** Implementing systems for establishing, communicating, monitoring, and enforcing social norms or rules within an AI agent population.

   - *Purpose:* To regulate agent behaviour, resolve conflicts, and maintain stability in artificial social systems.
   - *Implementation:* Developing reputation systems, voting mechanisms for rule adoption, or automated sanctioning for norm violations.
   - *Measurement:* Rates of norm adherence, conflict frequency and resolution success, fairness of resource distribution, emergence of stable cooperative patterns.

3. **Communication and Coordination Protocols:** Establishing standardized ways for agents to exchange information, signal intent, and coordinate actions.

   - *Purpose:* To enable effective collaboration and reduce misunderstandings or conflicts arising from poor communication.
   - *Implementation:* Defining shared languages, ontologies, or interaction protocols (e.g., contract nets, auctions for task allocation).
   - *Measurement:* Task completion rates in cooperative settings, bandwidth usage, ambiguity in communication, success of joint plan execution.

**Interactions and AI Control Contribution:** This layer builds upon the characteristics of individual agents (Layers 6-8) and examines their collective dynamics. Emergent phenomena at this layer can be highly complex and unpredictable, potentially posing unique safety risks (e.g., collusive behaviour, unintended escalations). Controls aim to shape the interaction environment or internal agent decision-making to promote positive sum outcomes. Limitations include the high dimensionality and non-linearity of multi-agent systems, making formal analysis difficult, and the challenge of predicting emergent behaviours that are not explicitly designed for.

## D.10   Layer 10: Societal Impact

**Role and Purpose:** This outermost layer considers the broad, long-term effects of AI deployment on society, including economic, cultural, ethical, and environmental impacts. Its purpose is to guide the overall trajectory of AI development and deployment in a way that aligns with societal well-being, justice, and sustainability.

**Measurement:** Economic indicators (job creation/displacement, productivity growth, GDP impact), social metrics (inequality indices, public trust in AI, measures of bias in AI-driven decisions), environmental footprint (energy consumption, CO2 emissions from AI), reports of significant AI-related incidents or harms, and compliance with legal and ethical frameworks.

**Control Methods:** (Controls at this layer are predominantly human-driven societal and governance mechanisms, rather than direct technical controls on AI systems themselves.)

1. **Legislation, Regulation, and Policy:** Governments and international bodies establishing laws, standards, and policies governing the development, deployment, and use of AI.

   - *Purpose:* To set legal boundaries, ensure accountability, protect fundamental rights, and mitigate large-scale risks.
   - *Implementation:* Enacting AI-specific laws (e.g., EU AI Act), creating regulatory agencies, mandating risk assessments or audits for certain AI applications.

 • *Measurement:* Rates of compliance with regulations, effectiveness in preventing harms, impact on innovation, public and expert feedback on policy.

2. **Ethical Guidelines, Codes of Conduct, and Industry Standards:** Professional organizations, research institutions, and industry consortia developing and promoting principles for responsible AI.

   • *Purpose:* To foster a culture of responsibility and guide AI practitioners in making ethical design and deployment choices.

   • *Implementation:* Publishing ethical frameworks (e.g., Asilomar AI Principles), developing best practice guides, creating certification programs.

   • *Measurement:* Adoption rates of guidelines, self-reported adherence, impact on AI system design and behaviour.

3. **Public Discourse, Education, and Stakeholder Engagement:** Efforts to inform the public about AI, foster critical discussion, and involve diverse stakeholders in shaping AI's future.

   • *Purpose:* To ensure that AI development is democratically accountable and reflects a broad range of societal values and concerns.

   • *Implementation:* Public education campaigns, media reporting, multi-stakeholder forums, citizen assemblies.

   • *Measurement:* Levels of public understanding and trust in AI, diversity of voices in AI governance debates, responsiveness of AI development to public concerns.

4. **Impact Assessments and Auditing Mechanisms:** Formal processes for evaluating the potential societal impacts of AI systems before and during deployment, and for independently verifying claims about their safety or fairness.

   • *Purpose:* To proactively identify, assess, and mitigate potential negative consequences of AI technologies.

   • *Implementation:* Mandating AI Impact Assessments (AIAs), establishing independent AI auditing bodies or standards.

   • *Measurement:* Quality and comprehensiveness of impact assessments, number of identified risks mitigated, transparency of audit processes.

**Interactions and AI Control Contribution:** This layer provides the ultimate context and oversight for all other layers. Societal values, translated into laws, regulations, and ethical norms, impose constraints and directives that ideally propagate downwards, influencing how AI is designed, built, trained, and deployed at every technical level. Controls at this layer are primarily directive and preventive at a macro scale. Limitations include the slow pace of legal and societal adaptation compared to technological change, the challenges of global coordination, political complexities, and the difficulty of perfectly forecasting and balancing diverse societal impacts.

**D.11  Interactions, AI Control Advancement, and the Hierarchical Approach**

The Alignment Control Stack illustrates that AI control is not a monolithic problem but a series of interlinked challenges and opportunities at different levels of abstraction. Lower layers (1-4) provide the physical and computational substrate; errors or vulnerabilities here can catastrophically affect all higher-level functions. Mid-layers (5-8) are where the AI's intelligence and alignment are primarily shaped and evaluated. Upper layers (9-10) deal with emergent multi-agent behaviours and the broad societal context.

This hierarchical framework advances AI control by:

• **Specifying Control Loci:** It clearly maps out where different types of controls can and should be implemented, from hardware safeguards (Layer 1) to ethical oversight (Layer 10). This allows for a defense-in-depth strategy.

• Control Types: Controls vary across the stack:

   – *Preventive controls* aim to stop undesirable events from occurring (e.g., secure hardware design in Layer 1, robust training data in Layer 5, ethical guidelines in Layer 10).

   – *Detective controls* aim to identify undesirable events when they occur (e.g., intrusion detection in Layer 2, output monitoring in Layer 6, auditing in Layer 10).

- *Corrective controls* aim to remedy undesirable events or their impacts (e.g., ECC in Layer 1, patch management in Layer 2, output filtering in Layer 6, model editing in Layer 7).
- *Directive controls* aim to guide behaviour towards desired outcomes (e.g., objective functions in Layer 5, RLHF in Layer 8, regulations in Layer 10).

- **Highlighting Limitations:** Each layer's controls have inherent limitations. Hardware controls (Layer 1) cannot fix flawed software logic (Layers 3-5). Output filters (Layer 6) are reactive and can be bypassed by novel malicious outputs. Even sophisticated alignment techniques (Layer 8) may face challenges with underspecified objectives or emergent properties. Societal controls (Layer 10) can be slow and incomplete. Recognizing these limitations is crucial for realistic risk assessment.

- **Structuring Implementation and Measurement:** The stack provides a framework for how controls are implemented (e.g., specific algorithms, hardware mechanisms, policy documents) and how their effectiveness is measured (e.g., performance metrics, error rates, compliance levels, audit findings), often involving feedback loops where measurements inform adjustments to controls.

Adopting a hierarchical approach, common in fields like network engineering (OSI model) or industrial control systems, offers significant advantages. It allows for modularity, where controls at one layer can be designed and analysed with some abstraction of the layers below. This facilitates the study and testing of interrelationships between different control layers. For example, one can investigate how choices in model architecture (Layer 4) impact the effectiveness of safety filters (Layer 6), or how interpretability findings (Layer 7) can improve reward modelling (Layer 8). By decomposing the complex problem of AI control into more manageable, interconnected layers, we can develop a more systematic, rigorous, and ultimately more effective strategy for ensuring AI systems are safe, aligned, and beneficial. This structured perspective is essential for applying formal control theory effectively, as it helps identify the relevant system dynamics, control inputs, and performance objectives at each level of abstraction.

Table 1: -Stack AI Control and Measurement Taxonomy

| Control Layer | Category | Focus | Measurements | Control Methods |
|---|---|---|---|---|
| Layer 1: Physical Infrastructure | Hardware Components | **Silicon/Computational Substrates** | Transistor-level measurements (voltage, current, switching speed); Temperature, power dissipation, electromagnetic interference | Voltage regulation, thermal management, error correction codes |
| | | **Specialized Processors (GPUs, TPUs, NPUs, FPGAs)** | FLOP/s capacity, memory bandwidth, cache hierarchies; Parallelization efficiency, utilization rates | Dynamic voltage/frequency scaling, workload scheduling |
| | Accelerator Architecture | **Specialized Processors (GPUs, TPUs, NPUs, FPGAs)** | FLOP/s capacity, memory bandwidth, cache hierarchies; Parallelization efficiency, utilization rates | Dynamic voltage/frequency scaling, workload scheduling |
| | Networking Infrastructure | **Interconnects and Communication** | Bandwidth, latency, packet loss rates; Network topology effects on distributed training | Traffic shaping, load balancing, congestion control |
| | Storage Systems | **Data Storage and Access** | Read/write speeds, storage capacity, data integrity; Cache hit ratios, storage hierarchy optimisation | Prefetching algorithms, data placement strategies |
| Layer 2: System Software | Operating System Kernel | **Resource Management** | CPU scheduling, memory allocation, I/O management; Process isolation, security boundaries | Priority scheduling, resource quotas, container limits |
| | Runtime Environments | **Execution Frameworks** | JVM, Python interpreter, Docker containers; Garbage collection, memory management | Heap size limits, execution timeouts, sandboxing |
| | Distributed Computing Layer | **Cluster Management** | Kubernetes orchestration, service meshes; Load balancing, fault tolerance, auto-scaling | Circuit breakers, retry policies, bulkhead patterns |

| Control Layer | Category | Focus | Measurements | Control Methods |
|---|---|---|---|---|
| Layer 3: AI Framework | Deep Learning Frameworks | **Core Computation Engines** | TensorFlow, PyTorch, JAX execution graphs; Automatic differentiation, optimizer states | Gradient clipping, learning rate scheduling, early stopping |
| | Training Infrastructure | **Distributed Training Systems** | Parameter servers, all-reduce algorithms; Data parallelism, model parallelism strategies | Synchronization barriers, gradient aggregation rules |
| | Inference Serving | **Model Deployment Platforms** | TensorFlow Serving, TorchServe, ONNX Runtime; Batching, caching, model versioning | Request queuing, response timeouts, model swapping |
| Layer 4: Model Architecture | Network Topology | **Architectural Patterns** | Transformer blocks, convolutional layers, recurrent units; Skip connections, attention mechanisms, normalisation layers | Architecture search, pruning, quantization |
| | Parameter Spaces | **Weight and Bias Distributions** | Parameter counts, sparsity patterns, quantization levels; Gradient flows, activation statistics | Weight initialization, regularization, dropout |
| | Computational Graphs | **Execution Patterns** | Forward/backward pass efficiency, memory usage; Operator fusion, graph optimisation | Graph rewriting, memory optimisation, operator scheduling |
| Layer 5: Training Process | Data Pipeline | **Data Processing and Augmentation** | Data loading, preprocessing, augmentation strategies; Batch construction, shuffle patterns | Data validation, quality filtering, corruption detection |
| | Optimisation Dynamics | **Learning Algorithms** | SGD, Adam, RMSprop parameter updates; Learning rate schedules, momentum terms | Adaptive learning rates, gradient scaling, loss smoothing |

| Control Layer | Category | Focus | Measurements | Control Methods |
|---|---|---|---|---|
| | Training Monitoring | **Progress Tracking** | Loss curves, metric tracking, checkpoint management; Hyperparameter logging, experiment tracking | Early stopping, checkpoint restoration, hyperparameter tuning |
| Layer 6: Behavioural Output | Task Performance | **Direct Measurements** | Accuracy, F1-scores, perplexity, BLEU scores; Latency, throughput, resource efficiency | Performance thresholds, output filtering, response validation |
| | Robustness Characteristics | **Reliability Under Stress** | Adversarial robustness, distribution shift handling; Noise tolerance, corruption resilience | Adversarial training, input sanitization, confidence thresholding |
| | Calibration and Uncertainty | **Confidence Estimation** | Prediction confidence, uncertainty quantification; Calibration error, out-of-distribution detection | Temperature scaling, uncertainty-based rejection |
| Layer 7: Interpretability and Explanation | Feature Attribution | **Internal State Analysis** | Gradient-based attribution (SHAP, LIME, Integrated Gradients); Activation patterns, attention visualizations | Attribution consistency checks, explanation validation |
| | Mechanistic Understanding | **Circuit-Level Analysis** | Neuron functionality, feature decomposition; Causal intervention experiments | Targeted ablations, activation steering, circuit disruption |
| | Human-Interpretable Outputs | **Explanation Generation** | Natural language explanations, visual saliency maps; Counterfactual examples, rule extraction | Explanation quality metrics, consistency enforcement |
| Layer 8: Preference and Reward | Value Alignment | **Goal Specification and Adherence** | Reward model fidelity, preference learning; Constitutional AI principles, value learning | Reward modelling, preference elicitation, value iteration |
| | Harm Prevention | **Risk Mitigation** | Toxicity detection, bias measurement; Safety classifier outputs, content filtering | Safety classifiers, output sanitization, risk scoring |

| Control Layer | Category | Focus | Measurements | Control Methods |
|---|---|---|---|---|
| | Deception and Manipulation Detection | **Trustworthiness Assessment** | Consistency checking, truth verification; Manipulation attempt detection, intent inference | Truth verification systems, consistency enforcement |
| Layer 9: Multi-Agent and Social | Agent Interaction Dynamics | **Inter-Agent Communication** | Cooperation vs. competition metrics; Information sharing patterns, coordination efficiency | Communication protocols, incentive alignment, mechanism design |
| | Collective Intelligence | **Emergent Group Behaviours** | Swarm intelligence metrics, collective decision-making; Group performance vs. individual capabilities | Group formation rules, consensus mechanisms, coordination algorithms |
| | Human-AI Collaboration | **Joint Performance Systems** | Team effectiveness, trust calibration; Handoff protocols, shared mental models | Trust calibration, interface design, collaboration protocols |
| Layer 10: Societal Impact | Economic Effects | **Market and Labor Impacts** | Job displacement rates, productivity changes; Market concentration, competitive effects | Impact assessment, gradual deployment, retraining programs |
| | Social and Cultural Influence | **Behavioural and Belief Changes** | Opinion polarization, information bubble effects; Cultural bias propagation, representation fairness | Diversity requirements, bias testing, cultural validation |
| | Governance and Regulation | **Policy and Compliance** | Regulatory compliance, audit requirements; Transparency obligations, accountability measures | Compliance monitoring, audit trails, regulatory reporting |

