# OpenReview forum: "Position: Out of Control - Why Alignment Needs Formal Control Theory"
_NeurIPS.cc/2025/Position_Paper_Track — Submitted to NeurIPS 2025 Position Paper Track_

### Official Review · Reviewer_nXQM · 2025-08-06

**Significance:** 3
**Presentation:** 4
**Rating:** 7
**Confidence:** 4

**Summary:**

The position paper argues that AI alignment research would benefit from formally adopting principles and tools from formal control theory. The authors identify two gaps in current alignment work: the formalization problem, where empirical or heuristic methods lack generalizable, rigorous guarantees; and the coordination problem, where alignment techniques across technical and socio-technical layers are developed in isolation. To address these, the authors propose the Alignment Control Stack (ACS), a ten-layer hierarchical taxonomy (from physical hardware up to societal governance) that specifies where control theory can be applied. The authors illustrate how formal control methods (e.g. Kalman filtering, stochastic optimal control, and game-theoretic formulations) can be mapped onto existing AI safety protocols, including auditing untrusted models and modeling adversarial subversion. Through toy simulations, they demonstrate deriving optimal adversary strategies, conducting sensitivity analyses, and framing adaptive deployment as POMDPs. The paper calls for integrating formal control frameworks with alignment methods to provide robust, interoperable guarantees essential for trustworthy AI deployment.

**Strengths:**

1. Clear Position & Motivation: The paper succinctly motivates why formal control theory’s emphasis on stability analysis, robust synthesis, and stochastic modeling complements existing alignment research.

2. Novel Taxonomy: The Alignment Control Stack provides a structured lens for organizing and comparing control interventions across layers, fostering clarity and interoperability.

3. Concrete Illustrations: By recasting recent AI safety protocols (e.g., auditing strategies, subversion games, adaptive deployment) in control-theoretic terms and showing toy simulations, the authors convincingly demonstrate how optimal control and game theory yield principled parameter choices and sensitivity insights.

4. Relevance & Impact: The call to bridge optimal control theory with AI alignment is timely for the NeurIPS community, given the increasing complexity of frontier models and the need for rigorous safety guarantees in high-stakes applications.

**Weaknesses:**

Though Section 5 discusses objections (complexity science, mechanistic interpretability, empirical iterative methods), the treatment is brief. For instance, more discussion of how white-box interpretability techniques could concretely feed into control models would strengthen interoperability claims.
The paper lacks experiments on realistic, high-dimensional models (e.g. large language models). The experiments focus on the variety of toy demonstrations of the Alignment Control Stack, but they do not yet overcome the core concerns about scalability to high-dimensional, real-world systems or about integrating human-in-the-loop dynamics and governance layers. The paper would still benefit from larger-scale case studies or at least mid-scale neural network experiments that validate tractability and human-agent coordination.

**Questions:**

How do you envision constructing tractable state-space models for large language models in Layers 5–7? Can you outline a concrete pipeline for extracting state variables (e.g., via probes or interpretability circuits) to feed into a stochastic control framework?

Have you evaluated approximate control methods (e.g., model predictive control with limited horizons) on mid-scale neural networks? What performance-safety trade-offs emerged?

Answering these questions would strengthen the argument made by the paper.

**Alternative Position:**

Yes, and alternative positions are well-considered and addressed by the argument

**Author Identification:**

No.

**Context:**

3

**Discussion:**

4

**Ethics:**

["NO or VERY MINOR ethics concerns only"]

**Position:**

Yes, the paper argues for or against a position related to machine learning.

**Support:**

3

**Thoroughness:**

4

---

### Official Review · Reviewer_AvWV · 2025-08-11

**Significance:** 4
**Presentation:** 2
**Rating:** 5
**Confidence:** 3

**Summary:**

The paper posits that formal optimal control theory should be central to AI alignment research, offering a distinct perspective from existing AI safety/security approaches. It identifies lack of formalisation and coordination as the two main challenges of alignments which can be addressed by control theory. To this end, the paper suggests Alignment Control Stack, a ten-layer hierarchical model spanning from physical hardware to socio-technical governance.

**Strengths:**

- The paper provides extensive reasoning for why formal control theory is necessary for alignment research which is quite relevant and topical.

- The hierarchal Alignment Control Stack helped ground the discussion on where to include control theory at every stage.

- The alternative views are well argumented.

- Connecting to governments and policy making is quite relevant and important.

**Weaknesses:**

- My biggest concern is which definition of mis-alignment was used? There are different definitions and they can be addressed with different methodologies. I suppose the paper wants to propose control theory to avoid misalignment in FM, but what are the misalignments that the authors are concerned with?

- Many examples use linear-Gaussian surrogates but these are quite different from the true nonlinear, high-dimensional, partially observable dynamics of frontier LLMs. Potentially these methods aren't easily applicable.

- There exists papers on aligment formalisation which the paper doesnt really discuss or acknowledge -  "Value alignment: a formal approach", "Artificial intelligence, values, and alignment"

**Questions:**

- I am not an expert in control theory, but do you think we can a hybrid approach with some methods from empirical research along with control theory can be realistically used. At what stack level which methods should be prioritised?

**Alternative Position:**

Yes, and alternative positions are well-considered and addressed by the argument

**Author Identification:**

No.

**Context:**

2

**Discussion:**

4

**Ethics:**

["NO or VERY MINOR ethics concerns only"]

**Position:**

Yes, the paper argues for or against a position related to machine learning.

**Support:**

3

**Thoroughness:**

3

---

### Official Review · Reviewer_7ea9 · 2025-08-12

**Significance:** 4
**Presentation:** 4
**Rating:** 9
**Confidence:** 4

**Summary:**

The paper argues that AI alignment research should integrate formal optimal control theory to address two issues:
(1) the formalisation problem: a lack of mathematical frameworks to generalize results and compare alignment methods
(2) the coordination problem: absence of a structured taxonomy for where in the AI system stack alignment interventions are applied.

It presents the Alignment Control Stack (ACS), a 10-layer hierarchical model from physical hardware to socio-technical governance. The authors demonstrate how formal control methods can enhance current AI safety protocols by providing provable guarantees, robustness against adversarial strategies, and principled trade-off analysis.

The position advocated is that bridging current empirically-driven AI safety methods with control theory’s proven mathematical frameworks will result in more generalizable, and trustworthy alignment strategies for advanced and agentic AI systems. The Alignment Control Stack’s layered-control approach appears to be a new contribution to the AI safety literature.

**Strengths:**

- The paper argues its position clearly and discusses the alternative views in detail.
- The paper presents a clear, well-structured Alignment Control Stack that organizes alignment interventions across ten hierarchical layers from hardware to socio-technical governance.
- The inclusion of simulated case studies strengthens its claims by showing how tools of control theory can provide strategic insights and formal guarantees.
- The paper is well-written, and the appendix provided more details into the stack.

**Weaknesses:**

- The paper lacks real-world empirical validation of the Alignment Control Stack beyond toy simulations.
- It provides limited implementation details on how to practically integrate control-theoretic methods into existing AI development pipelines.

Minor:
- There is no need for subsection 1.1 if it is only one sub-section
- There are a few typos in the paper, I suggest revising.

**Questions:**

- How feasible is constructing tractable formal models for high-dimensional complex AI systems like LLMs or agentic AI where internal dynamics are only partially understood and change with fine-tuning or external tool calling?

- In the Alignment Control Stack, how would you manage interactions across layers when control resources are limited, and could your framework quantify the balance or trade-off between acting at lower (technical) versus higher (socio-technical) layers?

**Alternative Position:**

Yes, and alternative positions are well-considered and addressed by the argument

**Author Identification:**

No.

**Context:**

4

**Discussion:**

3

**Ethics:**

["NO or VERY MINOR ethics concerns only"]

**Position:**

Yes, the paper argues for or against a position related to machine learning.

**Support:**

4

**Thoroughness:**

3

---

### Note · Authors · 2025-08-24

**1-10 Additional Comments:**

It is great to see Neurips adopt a position paper track as normative debates have an important influence on technical research directions.

**1-11 Submit Again:**

Definitely yes

**1-1 Submission Process:**

5

**1-2 Next Year:**

The position track might benefit from different streams - this could enable participants at the conference to more easily connect across related projects. It would also make it easier to discern the SOTA research frontiers and most significant open questions from a position-paper perspective.

Also a very minor point, we have found both with Neurips and ICML position papers that often reviewers seek experiments that might be more in line with main-track papers. We are happy to undertake these but at least for other proceedigns, the CFPs tend to require position papers to not introduce new methods. There is always a balance in how experimental v normative a paper is, but it is something perhaps worth considering in terms of more guidance.

**1-3 Future Development:**

One idea might be for a position or similar track to set out specific research questions to be addressed and then have papers competitively submit in addressing that question. This might help with comparative benchmarks (and reviewing) across what is no doubt a very diverse track.

**1-4 Interest:**

["Panel discussions with other position paper authors", "Structured debates on controversial topics", "Workshops for developing position papers", "Mentorship programs for early-career researchers"]

**1-5 Thoughtful:**

10

**1-6 Supportive:**

10

**1-7 Technical Aspects Versus Position:**

8

**1-8 Gate Keeping:**

5

**1-9 Camera Ready Changes:**

Each reviewer provided helpful and constructive feedback, we will be integrating their changes to improve the manuscript. We have set out the changes we will make in response to the reviewers below but paraphrase them for completeness:

1. Clarify definition of misalignment as divergence between intended objectives and realised behaviour, spanning technical, behavioural, and socio-technical domains.

2. Correct typos, simplify subsection structure, and standardise terminology/figure labels and include explicit references to prior alignment-formalisation work mentioned in the review.

3. Include discussion of how ACS could be applied to open-weights models across layers.

4. Provide greater detail on measurement and control signals per ACS layer.

5. Expand discussion of surrogate modelling approaches (reduced-order models from telemetry, probes, activation pathways).

6. Add notes on approximate control methods (e.g. MPC, short horizon) as future research pathways.

7. Expanded Section 5 with deeper treatment of objections and alternatives, especially mechanistic interpretability and complexity-science perspectives.

8. Mention hybrid approaches as a synthesis of empirical methods with control theory, noting which ACS layers are most suited to which.

9. Discussion of scalability limits, emphasising toy simulations and outlining pathways to mid-scale validation studies.

10. Sketch a framework in the Appendix for allocating limited control budget across ACS layers by comparing estimated impact vs. cost.

11. Clarify that linear-Gaussian surrogates are illustrative simplifications while real systems are nonlinear, high-dimensional, and partially observable.

12. Stronger link to implementation details via mapping ACS layers to practical ML devops processes.

13. Emphasising that ACS provides structure for interoperability, not a replacement for existing empirical safety work.

We will also streamline and simplify the empirical sections.

**3-1 Review Response1:**

7ea9

**3-2 Reaction To Review1:**

Thank you for the thoughtful review of our paper on the Alignement Control Stack (ACS). Your recommendations/corrections will assist us to improve the paper. We set out specific responses below.

1. We will include additional discussion in the Appendix of how ACS could be applied to open-weight models to exercise multiple layers (training adjustments, behavioural gating, interpretability steering).

2. We will include in the Appendix greater detail on specific measurement and control signals at different layers in the ACS.

Regarding the specific questions:

1. Exactly how to apply formal control principles to higher-dimensional complex models and where measurement/understanding of internal dynamics remains uncertain. We will include prospective examples of how this complexity may be addressed. One example includes the use of reduced-order surrogate models identified from telemetry (e.g., safety scores, calibration error, probes), which can be adaptively updated and used to enforce safety envelopes even under partial observability. Another example we will flag includes the use of robust metrology to identify persistent and representative (or computationally faithful) AI observables at different layers (e.g. at the circuit level, or behavioural level) and then using experimental probes to conduct ablation studies for how robust such observables are. The analogy here is with statistical physics, where control may be exerted at a 'macroscopic' or intermediate level in a system while microscopic (atomic) trajectories remain individually uncontrolled.

2. This is an excellent point. In relation to resource-constrained interoperability across ACS layers, we will include a discussion on resource-bounded prioritisation: when control budget is limited, interventions can be allocated across layers by comparing estimated impact vs. cost using empirical methods. This provides a principled way to balance broad, expensive low-level controls with targeted high-level measures.

**3-3 Review Response2:**

AvWV

**3-4 Reaction To Review2:**

Thank you for the constructive review of our paper on the Alignment Control Stack (ACS). Your comments will assist us to improve the manuscript. Regarding your specific comments and questions:

1. We will clarify that by misalignment we mean divergence between intended objectives (human- or regulator-specified) and realised system behaviour, spanning technical (e.g., specification gaming, goal misgeneralisation), behavioural (unsafe or deceptive outputs), and socio-technical harms (misinformation, misuse). The ACS is intended to map where such divergences can be measured and controlled.

2. We acknowledge that linear/gaussian surrogates these are simplifications. This is quite common in control theory (where often the aim is to reduce non-linear complexity to tractable, if imperfect, linear approximations. For our purposes, these are illustrative toy models to demonstrate how formal control concepts (estimation, stability, robust synthesis) apply. We will emphasise that true LLM dynamics are nonlinear, high-dimensional, and partially observable and discuss how control theory may nevertheless provide techniques to address this.

3. We will make specific reference to the two references as examples in the literature. The "Value alignment" focuses upon alignment of preferences, while the "Artificial intelligence, values, and alignment" paper is more philosophical. Our approach aims for a wider technical synthesis with formal control theory beyond these specific applications.

4. Your suggestion of hybrid approahces is exactly what we would recommend and we will emphasise this point i.e. a synthesis of theory and empirical approaches. As to where in the stack, we suspect there is no universal rule and it depends on the state of technology, but that this will call for empirical methods/evaluations that can identify where/how in the ACS particular models are most suitably controlled and aligned.

**3-5 Review Response3:**

nXQM

**3-6 Reaction To Review3:**

Thank you for your helpful review. Your comments and questions will assist us in improving the paper.

Addressing your questions/comments specifically:

1. We agree Section 5 ought to be extended. We will expand the discussion of current proposals from mechanistic interpretability (e.g., probes, circuit discovery) are being proposed to inform state variable selection for control models. We will also emphasise empirical and complexity-science as important complements to the use of formal control theory.

2. We wil discuss how larger-scale experiments may be run to validate formal control methods. As this paper is a position paper, we kept the empirical/simulations limited. However, your point and question is central to the useful ability to control such systems and so we will include greater discussion of it in the main body of the paper and Appendix.

On the specific questions:

1. We intend to set out how surrogate modelling via probes or interpretability circuits may be used (and are being used) to extract low-dimensional, behaviour-relevant state space models using (e.g., safety scores, calibration errors, activation pathways). We will include discussion in the Appendix on when and how surrogates may provide approximate state variables suitable for stochastic control frameworks (e.g. when does a surrogate model provide guarantees for higher-order models, something control theory does address). The other approach is to take a measurement-theoretic approach to identify AI observables at different scales akin to what you see in control via statistical mechanical principles (e.g. you don't control all the atomic interactions, just the mesoscopic/macroscopic behaviour).

2. We have not yet evaluated MPC methods/approximate control fully as yet. The paper was the first in a series of control theory-meets-mechanistic interpretability papers that we are undertaking. The suggestions are excellent and we will note them as pathways for future research in the paper.

---

### Meta-Review · Area_Chair_vuHi · 2025-09-12

**Rating:** 8
**Confidence:** 4

**Strengths:**

The paper addresses a highly significant and timely challenge of formalizing AI alignment and integrating it with control theory. The Alignment Control Stack (ACS) is highlighted as a novel, structured taxonomy that organizes alignment interventions across technical and socio-technical layers. The paper is clear, well-written, and connects ML safety with broader governance and policy contexts. Alternative positions are discussed, and the work is positioned to have a broad impact.

**Weaknesses:**

The main limitations are around concreteness and scalability. The framework is demonstrated primarily with toy models, leaving open questions about tractability and applicability to larger LLMs or agentic AI. Some reviewers noted limited discussion of existing formal alignment literature and a need for more integration with interpretability methods.

**Questions:**

How feasible is it to build tractable formal models for high-dimensional, partially observable systems like LLMs?

**Ethics:**

There are no major ethical concerns.

**Thoroughness:**

4

---

### Decision · Program_Chairs · 2025-09-26

Reject